# Targeted Unlearning via Single Layer Unlearning Gradient

## Abstract

The unauthorized generation of privacy-related and copyright-infringing content using generative-AI is becoming a significant concern for society, raising ethical, legal, and privacy issues that demand urgent attention. Recently, machine unlearning techniques have arisen that attempt to eliminate the influence of sensitive content used during model training, but they often require extensive updates in the model, reduce the utility of the models for unrelated content, and/or incur substantial computational costs. In this work, we propose a novel and efficient method called Single Layer Unlearning Gradient (SLUG), that can unlearn targeted information by updating a single targeted layer of a model using a one-time gradient computation. We introduce two metrics: layer importance and gradient alignment, to identify the appropriate layers for unlearning targeted information. Our method is highly modular and enables selective removal of multiple concepts from the generated outputs of widely used foundation models (e.g., CLIP), generative models (e.g., Stable Diffusion) and Vision-Language models. Our method shows effectiveness on a broad spectrum of concepts ranging from concrete (e.g., celebrity name, intellectual property figure, and object) to abstract (e.g., novel concept and artistic style). Our method also exhibits state-of-the-art efficiency with effective unlearning and retention on the comprehensive benchmark *UnlearnCanvas*. Our code is available at https://anonymous.4open.science/r/SLUG-6CDF.

## 1 Introduction

Modern machine learning models, including large language models (LLMs) (Achiam et al., 2023; Leiter et al., 2024), Stable Diffusion (SD) (Salimans & Ho, 2022; Yang et al., 2023) , and vision language mdoels (VLMs) (Zhang et al., 2024b; Liu et al., 2024a) leverage vast amounts of data for training. While these large unsupervised datasets enhance performance under scaling law (Kaplan et al., 2020), they also raise serious data privacy and legal compliance (gdp, 2016; Thiel, 2023) concerns as sensitive, unsafe, and unwanted data can influence the trained models (Thiel, 2023). Completely abandoning trained model weights and re-training large models from scratch using scrutinized dataset is prohibitively expensive and wasteful. Machine unlearning (Cao & Yang, 2015; Nguyen et al., 2022) is an attractive alternative, which refers to a broad set of techniques designed to reverse the learning process, with the specific aim to efficiently remove targeted information from a trained model without re-training the model from scratch.

Machine unlearning has three main objectives: **(1) Low computational cost**, as the naïve approach of re-training models usually achieves the best result (exact unlearning) at the expense of large computational cost. **(2) Effective unlearning**, to ensure that the model forgets the intended data completely. **(3) Utility retention**, maintaining the original model performance, in terms of accuracy and utility on data/tasks unrelated to the intended unlearning data. Current unlearning methods often fall short of meeting all these objectives simultaneously. Traditional approaches like fine-tuning (FT) (Warnecke et al., 2023) and gradient ascent (GA) (Thudi et al., 2022) struggle to balance effective forgetting with utility preservation. More recent techniques such as saliency unlearning (SalUn) (Fan et al., 2024) and selective synaptic dampening (SSD) (Foster et al., 2024) attempt to address this by

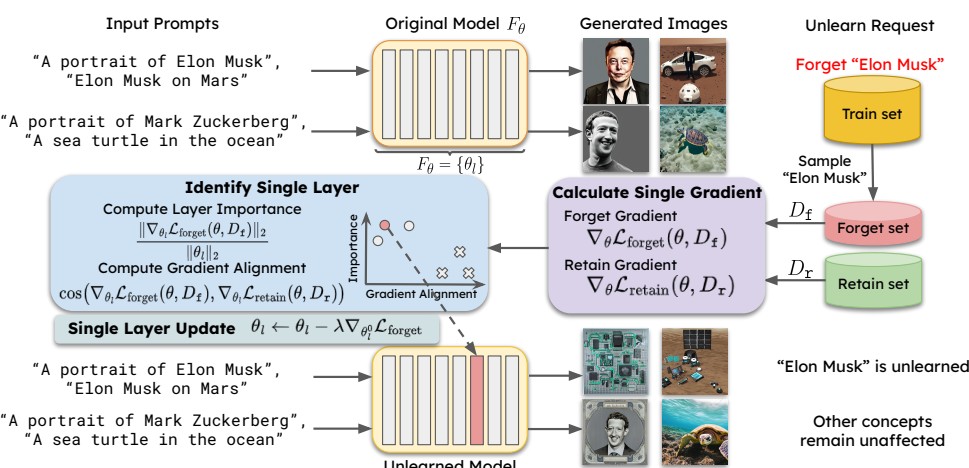

**Figure 1:** Overview of our proposed **S**ingle **L**ayer **U**nlearning **G**radient (SLUG) framework. Given an unlearning query, we curate a forget set and retain set, then compute corresponding gradients. The gradient alignment guide identifying single layer updates for unlearning. A binary search helps determine the step size $\lambda$, effectively erasing specified concepts while preserving the model's utility.

identifying and updating only salient parameters. While these methods improve overall unlearning performance, they still face the following challenges. First, they usually involve iterative updates over the model parameters, resulting in high computational costs (Fan et al., 2024). Second, the significant weights targeted for updates are often spread throughout the model, offering limited insight into the model structure. Finally, they require careful hyperparameter tuning, including learning rate, number of iterations, and parameters for selecting suitable masks in saliency approaches.

In this paper, we propose a novel and efficient method for targeted unlearning, namely **Single Layer Unlearning Gradient (SLUG)**. We push the boundaries of efficiency as our algorithm identifies and updates a single layer using a single gradient computation to achieves effective unlearning without affecting the general utility of the large pretrained models. Figure 1 provides an overview of our proposed framework. We first calculate gradients of forget and retain losses with respect to the model weights using a designated or curated forget and retain set. The forget and retain sets contain images associated with concepts that are targeted to be removed from, and retained in the model, respectively. Based on these gradients, we introduce two metrics — layer importance and gradient alignment, to identify the appropriate layers for unlearning targeted concepts. To determine a suitable step size for model weight updates, SLUG employs binary search along the direction of forget gradients. We demonstrate that SLUG outperforms state-of-the-art methods in unlearning models involving CLIP, and SD, across various tasks and architectures. We also evaluate SLUG on a comprehensive unlearning benchmark *UnlearnCanvas* Zhang et al. (2024c), showcase its superiority in efficiency and balancing trade-off between unlearning targeted concept and retaining model utility. In addition to its efficiency and effectiveness, our methods offers higher modularity and better interpretability compared to Fan et al. (2024); Foster et al. (2024). SLUG precisely identifies layers associated with distinct concepts, which provides insights into the features learned by different layers and their functionalities, offering generalized guidance for new tasks and model architectures design.

## 2 BACKGROUND

### 2.1 MACHINE UNLEARNING PRELIMINARIES

The goal of machine unlearning is to remove the influence of a specific subset of training data, $D_{\mathtt{f}} \subset D$, on a pre-trained model $F_\theta(D)$ with parameters $\theta$. The challenge is to make this process more efficient than re-training the model on the retain set $D_{\mathtt{r}} = D \setminus D_{\mathtt{f}}$. The unlearning algorithm $U$ should produce an unlearned model $F_{\theta_{\mathtt{f}}} = U(F_\theta(D), D, D_{\mathtt{f}})$ that is functionally equivalent to a model retrained only on $D_{\mathtt{r}}$, i.e., $F_{\theta_{\mathtt{r}}}(D_{\mathtt{r}})$. We can formulate the

unlearning problem as

$$\min_{\theta} \underbrace{\frac{1}{N_{\mathrm{r}}} \sum_{(x_{\mathrm{r}}, y_{\mathrm{r}}) \in D_{\mathrm{r}}} \ell(F_{\theta}(x_{\mathrm{r}}), y_{\mathrm{r}})}_{\mathcal{L}_{\mathrm{retain}}} - \underbrace{\frac{\alpha}{N_{\mathrm{f}}} \sum_{(x_{\mathrm{f}}, y_{\mathrm{f}}) \in D_{\mathrm{f}}} \ell(F_{\theta}(x_{\mathrm{f}}), y_{\mathrm{f}})}_{\mathcal{L}_{\mathrm{forget}}}, \tag{1}$$

where $N$ is the number of elements in $D$, $\alpha$ is a balancing factor, and $\ell$ is the loss function.

Naïve gradient ascent (GA) on the forget set increases forget loss but risks over-unlearning (i.e., reducing accuracy on the retain set). Fine-tuning (FT) on the retain set is poor at unlearning but can mitigate over-unlearning when combined with GA in a two-stage approach (Fan et al., 2024), which we call GAFT (equation 1).

Recent methods like SalUn and SSD (Fan et al., 2024; Foster et al., 2024) focus on updating only salient parameters, determined through gradient analysis, to stabilize unlearning. SalUn (Fan et al., 2024) applies hard thresholds on forget-loss gradients, while SSD (Foster et al., 2024) dampens important weights for both the forget and retain sets. Despite improving unlearning performance, these methods involve complex hyperparameter tuning and lack interpretability. This motivates us to develop a hyperparameter-free, interpretable method.

### 2.1.1 VISION LANGUAGE ALIGNMENT

Traditional machine unlearning approaches often struggle with high computational costs and limited scalability, which restricts their application to small-scale image classification models (Jia et al., 2023; Foster et al., 2024). In contrast, our method breaks away from these constraints by offering superior scalability and flexibility, making it suitable for large multi-modal foundation models such as CLIP (Radford et al., 2021), Stable Diffusion (Rombach et al., 2022), and vision language models (VLMs) (Liu et al., 2024a).

CLIP (Radford et al., 2021), in particular, is pivotal in advancing multi-modal models by aligning visual and textual representations through contrastive loss (Chopra et al., 2005):

$$\ell = \frac{1}{2N} \sum_{i=1}^{N} \left( \ell_{i2t}(i) + \ell_{t2i}(i) \right), \tag{2}$$

$$\ell_{i2t}(i) = -\log \frac{\exp(\cos(\mathbf{v}_i, \mathbf{t}_i)/\tau)}{\sum_{j=1}^{N} \exp(\cos(\mathbf{v}_i, \mathbf{t}_j)/\tau)}, \quad \ell_{t2i}(i) = -\log \frac{\exp(\cos(\mathbf{t}_i, \mathbf{v}_i)/\tau)}{\sum_{j=1}^{N} \exp(\cos(\mathbf{t}_i, \mathbf{v}_j)/\tau)}. \tag{3}$$

Here, $\mathbf{v}_i$ is the normalized image embedding from the vision model $f_{\mathrm{v}}$, and $\mathbf{t}_i$ is the normalized text embedding from the text model $f_{\mathrm{t}}$. The temperature $\tau$ controls the sharpness of the softmax probability distribution, while cosine similarity is defined as $\cos(\mathbf{v}_i, \mathbf{t}_j) = \mathbf{v}_i \cdot \mathbf{t}_j$. Minimizing this contrastive loss aligns the vision and language representations in the embedding space. In unlearning, one of our goals is to break these learned alignments.

### 2.1.2 LOSS FUNCTIONS FOR GRADIENT CALCULATION

Selection of an appropriate loss functions to perform unlearning is critical. For image classification models, cross-entropy loss can be directly used as both retain loss and forget loss. However, the scenario for contrastive learning is different. For the retain set, we can still use the original contrastive loss as in equation 2:

$$\mathcal{L}_{\mathrm{retain}} = \frac{1}{2N} \sum_{i=1}^{N} \left( \ell_{i2t}(i) + \ell_{t2i}(i) \right). \tag{4}$$

For the forget set, we employ the cosine embedding loss:

$$\mathcal{L}_{\mathrm{forget}}(\mathbf{v}_i, \mathbf{t}_j) = 1 - \cos(\mathbf{v}_i, \mathbf{t}_j). \tag{5}$$

This loss directly pushes the embeddings of positive pairs away while not tampering with the embeddings of negative pairs. Using the original contrastive loss as forget loss will result in ineffective unlearning.

## 3 SINGLE LAYER UNLEARNING GRADIENT (SLUG)

Our proposed approach SLUG performs three main steps using given unlearning query with retain and forget sets: (1) calculate one-time gradients for the forget and retain losses; (2) identify a single layer with high importance to the forget set and low relevance to the retain set; (3) update the targeted layer along a linear path using one-time calculated gradient. The framework is illustrated in Figure 1. Our approach improves the state-of-the-art along three axes: (1) low computational cost, (2) effective unlearning, and (3) high retention of general utility.

### 3.1 LAYER IDENTIFICATION

We are inspired by how different layers in deep networks learn distinct features; early layers capture basic patterns like edges, while later layers focus on more specific details (Zeiler & Fergus, 2014; Olah et al., 2017; Ghiasi et al., 2022). To efficiently unlearn, we aim to modify only those layers that directly hold the information related to the unlearning task, avoiding changes to layers processing abstract features unrelated to the data to be forgotten. Our goal is to identify the layers most critical to unlearning while preserving the model's overall functionality. We achieve this by performing unlearning within the "nullspace" of the retain set, focusing on layers that minimally impact retained data performance while effectively removing the targeted features. This approach improves the precision of unlearning and provides insights into how the model handles data retention and unlearning.

To measure the influence of each parameter, similar to (Aich, 2021; Foster et al., 2024), we use the Fisher information matrix(Kay, 1993; Hassibi et al., 1993; Kirkpatrick et al., 2017), approximated by its diagonal for simplicity:

$$\mathcal{I}_D(\theta) = -\mathbb{E}\left[\frac{\partial^2}{\partial\theta^2}\log L(\theta; D)\right] = \mathbb{E}\left[\left(\frac{\partial}{\partial\theta}\log L(\theta; D)\right)\left(\frac{\partial}{\partial\theta}\log L(\theta; D)\right)^{\mathsf{T}}\right]. \quad (6)$$

The diagonal elements reflect the sensitivity of the log-likelihood to parameter changes, and we extend this to layers by aggregating sensitivities. The importance of a layer is determined by the ratio of the $\ell_2$ norm of the forget loss gradients to the $\ell_2$ norm of the layer's parameters:

$$\texttt{Importance of layer l:}\ \mathrm{Importance}(l) = \frac{\sqrt{\mathcal{I}_{D_{\mathtt{f}}}(\theta_l)}}{\|\theta_l\|_2} = \frac{\|\nabla_{\theta_l}\mathcal{L}_{\mathrm{forget}}(\theta, D_{\mathtt{f}})\|_2}{\|\theta_l\|_2}. \quad (7)$$

Importance of layer alone is insufficient. We also ensure that forget gradients are nearly orthogonal to the retain gradients by minimizing the gradient alignment:

$$\texttt{Gradient alignment:}\ \mathrm{Alignment}(l) = \cos\bigl(\nabla_{\theta_l}\mathcal{L}_{\mathrm{forget}}(\theta, D_{\mathtt{f}}), \nabla_{\theta_l}\mathcal{L}_{\mathrm{retain}}(\theta, D_{\mathtt{r}})\bigr). \quad (8)$$

Small alignment would prevent unlearning updates from negatively affecting the retain set.

To balance both objectives, we use the concept of a Pareto optimal set (Marler & Arora, 2010), optimizing both importance and gradient orthogonality. Figure 2 illustrates the Pareto front for unlearning a person identity from CLIP ViT-B-32, showing layers (as colored dots) that achieve optimal trade-offs between these objectives, where improving one metric would necessarily worsen the other.

### 3.2 LINEARIZING UNLEARNING TRAJECTORY IN A SINGLE GRADIENT DIRECTION

Existing unlearning methods calculate gradients at each iteration to update model parameters, which significantly increases computational complexity. Inspired by task arithmetic (Ilharco et al., 2023) and the linear nature of many optimization problems (LeCun et al., 2015), we observe that repeated gradient calculations can be redundant. Instead, we propose calculating the gradient only once for the initial model and updating the parameters $\theta_l$ of any layer $l$ in a weight-arithmetic fashion. Specifically, the weights are updated along a fixed gradient direction:

$$\theta_l^* \leftarrow \theta_l^{(0)} - \lambda^* \nabla_{\theta_l}\mathcal{L}_{\mathrm{forget}}(\theta, D_{\mathtt{f}})\Big|_{\theta=\theta^{(0)}}, \quad (9)$$

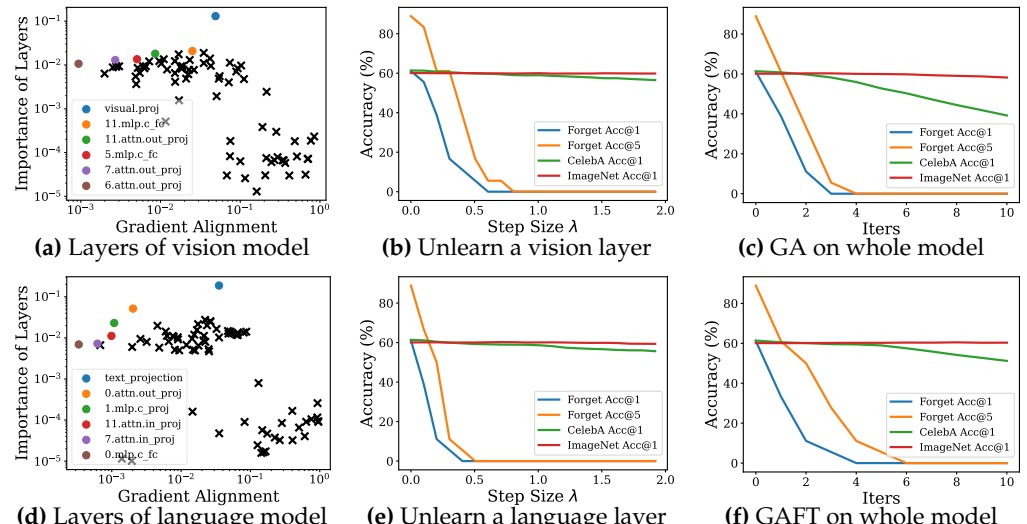

**(a)** Layers of vision model **(b)** Unlearn a vision layer **(c)** GA on whole model

**(d)** Layers of language model **(e)** Unlearn a language layer **(f)** GAFT on whole model

**Figure 2:** Layer identification (a,d) and unlearning with a single gradient (b,e). The first column shows gradient alignment and importance metrics for vision and language models from CLIP ViT-B-32, highlighting layers on the Pareto front for unlearning an identity. The second column demonstrates effective unlearning by updating identified layers along a single gradient direction without significantly impacting retain set performance. The third column shows that iterative methods (GA and GAFT) offer no advantage over a single gradient and require early stopping to prevent over-unlearning.

where $\theta_l^*$ represents the parameters of layer $l$ for the unlearned model and $\theta_l^{(0)}$ represents the initial parameters. The gradient $\nabla_{\theta_l} \mathcal{L}_{\text{forget}}(\theta, D_{\text{f}})\big|_{\theta=\theta^{(0)}}$ is calculated only once, based on the forget loss $\mathcal{L}_{\text{forget}}$ evaluated on the forget set $D_{\text{f}}$. The step size $\lambda^*$ controls the update magnitude.

Updating weights of a layer along a fixed gradient direction is equivalent to linearizing the unlearning trajectory. This approach reduces computational complexity while ensuring effective unlearning. To select the appropriate step size $\lambda^*$, we perform a binary search along the linearized path, halting when the evaluation metric indicates satisfactory unlearning without harming performance on the retain set. For example, we stop at $\lambda \approx 0.75$ in Figure 2b, where the forget accuracy is near zero and test accuracy is high. This method strikes a balance between computational efficiency and precision, maintaining model utility while achieving unlearning goals.

### 3.3 GENERALIZATION TO STABLE DIFFUSION AND VLMS

By harnessing effective unlearning in CLIP models, our technique can be extended to larger models built on CLIP, such as Stable Diffusion (Rombach et al., 2022; Salimans & Ho, 2022) and VLMs like LLaVA (Liu et al., 2024a; 2023).

**Unlearn Stable Diffusion**. Diffusion models, known for generating high-quality images from text, use a text encoder (e.g., CLIP ViT-H/14 in Stable Diffusion) to embed prompts into a high-dimensional space. The text embedding guides the denoising process through cross-attention, starting from an initial noise $\mathbf{x}_T$ and iteratively updating the noisy image at each step:

$$\mathbf{x}_{t-1} = \sqrt{\alpha_t} \left( \mathbf{x}_t - \gamma_t \nabla_\mathbf{x} \log p(\mathbf{x}_t|\mathbf{e}) \right) + \sqrt{1-\alpha_t} \mathbf{z}_t. \tag{10}$$

Here, $\mathbf{x}_t$ is the noisy image at step $t$, $\mathbf{z}_t$ is the noise added at step $t$, $\alpha_t$ is a time-dependent parameter controlling the noise balance, $\gamma_t$ is the learning rate, $\mathbf{e} = f_\mathbf{t}(\texttt{txt})$ is the text embedding, and $\nabla_\mathbf{x} \log p(\mathbf{x}_t|\mathbf{e})$ is the gradient of the log-probability of the noisy image given the text embedding, guiding the denoising process. We freeze the CLIP vision model and only update the language model to achieve unlearning.

**Unlearn VLMs**. Vision-Language Models (VLMs) enable LLMs to process multi-modal information. LLaVA-1.5 (Liu et al., 2023) uses a pretrained CLIP vision encoder ViT-L/14-336px to extract the visual features $\mathbf{e} = f_{\mathbf{v}}(\texttt{img})$, which are projected as visual tokens $\mathbf{H_v} = \mathbf{W} \cdot \mathbf{e}$ through an MLP $\mathbf{W}$. These tokens are then concatenated with language tokens $\mathbf{H_q}$ as input $\mathbf{H} = [\mathbf{H_v}; \mathbf{H_q}]$ to the language model. Since VLMs rely on the vision encoder, unlearning specific concepts in the CLIP vision model can directly influence the language model's output.

## 4 EXPERIMENTS AND RESULTS

### 4.1 EXPERIMENT SETUP

**Unlearning scenarios.** We investigate three main types of unlearning scenarios for practical and generalizable impact: (1) Unlearning identity information to counter privacy concerns; (2) Unlearning copyrighted content for compliance with legal standards. We primarily focus on large-scale multimodal models that include CLIP (Radford et al., 2021), Stable Diffusion (Rombach et al., 2022), and VLMs (Liu et al., 2024a); and (3) Unlearning artistic style and object concepts in *UnlearnCavas* Zhang et al. (2024c).

**Models.** We performed experiments on various models to demonstrate the broad applicability of our unlearning method. For CLIP models, we used architectures ranging from ViT-B-32 to EVA01-g-14, trained on LAION-400M dataset (Schuhmann et al., 2021), and model weights sourced from the OpenCLIP repository (Cherti et al., 2023). For Stable Diffusion models, we used the latest version from StabilityAI, which incorporates the CLIP-ViT-H-14 trained on the LAION-5B dataset. For VLMs, we used the improved LLaVA-v1.5 model from HuggingFace, which employs a CLIP ViT-L/14-336px model, trained by OpenAI, as the visual extractor.

**Datasets.** We used publicly-available datasets to construct the forget, retain, and evaluation sets. For unlearning target identities, we curated the forget set by filtering the LAION-400M dataset to isolate 1,000 to 6,000 image-text pairs per identity. The retain set consists of a single shard from LAION-400M, containing approximately 7,900 images (due to expiring URLs). To assess unlearning effectiveness, we used the CelebA dataset (Liu et al., 2015), sampling 100 frequently appearing celebrities from LAION-400M. Post-unlearning, model utility was evaluated using the ImageNet dataset for zero-shot classification. *UnlearnCanvas* Zhang et al. (2024c) was used to test unlearning of artistic styles and objects in Stable Diffusion.

**Evaluation metrics.** For CLIP models, we measure unlearning performance using forget accuracy, defined as the zero-shot classification accuracy on unlearned content. Following the standard zero-shot paradigm (Radford et al., 2021), predictions are based on the highest cosine similarity between image and text embeddings. Utility is assessed via zero-shot accuracy on ImageNet and CelebA. In addition to quantitative results for CLIP, we provide qualitative results from Stable Diffusion (image generation) and VLMs (question-answering) before and after unlearning. The UnlearnCanvas benchmark evaluates unlearning using diverse metrics, including computation and storage efficiency.

**Comparing methods.** We compare with the state-of-the-art methods along with classical methods. For unlearning in CLIP models, we compare with classical fine tuning (FT) (Warnecke et al., 2023), gradient ascent (GA) / negative gradient (NG) (Thudi et al., 2022), and recent salient parameters based saliency unlearning (SalUn) (Fan et al., 2024), and selective synaptic dampening (SSD) (Foster et al., 2024). For unlearning in Stable Diffusion models, we compare with 9 methods in Table 2 included in *UnlearnCanvas*.

### 4.2 UNLEARNING FOR CLIP

**Unlearning identities.** We demonstrate that modifying a single layer suffices to unlearn an identity or concept while preserving the model's overall utility. Figure 3 presents an example of unlearning identity for Elon Musk images. Each cell in these matrices shows the cosine similarity between the embeddings of an image-text pair. Before unlearning (Figure 3a), we observe high similarity (bright spots) along the diagonal, indicating strong

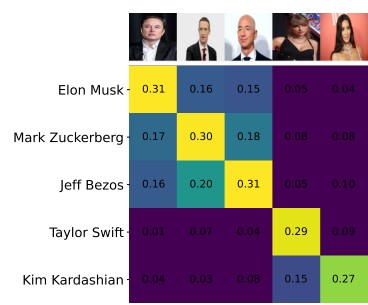

**(a)** Original cosine similarity matrix



**(b)** Cosine similarity matrix after unlearning

**Figure 3:** Cosine similarity matrix of image-text pairs before & after unlearning "Elon Musk" as an example. (a) original CLIP correctly associate images and text of distinct identities with high similarity. (b) after unlearning, the image-text pair of "Elon Musk" is no longer matched, while other identities are only slightly affected.

**Table 1:** Performance comparison of different unlearning methods on CLIP zero-shot classification. FA@{1, 5} stands for top-{1, 5} forget accuracy (%), i.e., accuracy of unlearned identity. TA_IN@1 and TA_CA@1 stands for the top-1 test accuracy (%) on ImageNet and CelebA dataset, respectively. $K$ and $k$ denotes the number of epochs for training and iterations for unlearning, respectively ($K = 32$ and $k = 10$ in our experiments). $N$ is the size of whole training set, which is much larger than our sampled forget set ($N_f$) and retain set ($N_r$). We report results for two learning rates. Best performing results are highlighted in red color.

| Method | FA@1 ($\downarrow$) | FA@5 ($\downarrow$) | TA_IN@1 ($\uparrow$) | TA_CA@1 ($\uparrow$) | Compute Time ($\mathcal{O}$) |
|---|---|---|---|---|---|
| Original | 73.05 | 92.22 | 60.12 | 61.38 | $\mathcal{O}(K \cdot N)$ |
| | | | learning rate $= 10^{-6}$ / $10^{-7}$ | | |
| FT (Warnecke et al., 2023) | 66.08/70.50 | 90.10/92.22 | **60.36**/60.26 | 60.70/**61.35** | $\mathcal{O}(k \cdot N_r)$ |
| GA (Thudi et al., 2022) | **0.00**/0.00 | **0.00**/4.91 | 35.88/60.03 | 24.92/53.86 | $\mathcal{O}(k \cdot N_f)$ |
| GAFT (equation 1) | **0.00**/2.67 | **0.00**/15.89 | 55.52/60.13 | 25.71/55.22 | $\mathcal{O}(k \cdot (N_f + N_r))$ |
| SalUn (Fan et al., 2024) | **0.00**/3.33 | **0.00**/15.69 | 55.45/60.26 | 26.11/55.81 | $\mathcal{O}(N_f) + \mathcal{O}(k \cdot (N_f + N_r))$ |
| SSD (Foster et al., 2024) | **0.00** | **0.00** | 51.84 | 35.96 | $\mathcal{O}(N_f + N_r)$ |
| SLUG (ours) | **0.00** | **0.00** | 59.96 | 58.32 | $\mathcal{O}(N_f + N_r)$ |

alignment between images and their corresponding text descriptions across all identities. After unlearning Elon Musk (Figure 3b), we see a marked decrease in similarity for the Elon Musk image-text pairs (visible as a darkened region), while other identities remain largely unaffected. This demonstrates our method's precision in selectively removing specific information. Similar results for additional identities and model architectures are presented in Figure 6 in the Appendix, further supporting the generalizability of our approach. Moreover, Figure 8 in the Appendix showcases our method's capability to simultaneously unlearn multiple identities, highlighting its scalability and flexibility.

**Unlearning without losing utility.** One noteworthy attribute of our method is that the performance on unrelated tasks, like ImageNet for common object recognition, remains intact. Table 1 presents the quantitative performance and comparison of different methods for classification with ImageNet and CelebA. Note that for closely related tasks such as CelebA, which focuses on face recognition, there is a slight impact on performance. As shown in Table 1, our method outperforms other comparing methods in terms of forget and retain accuracy. Furthermore, the overall computational complexity of our method is minimal as it computes a one-time gradient (for forget and retain set, thus $\mathcal{O}(N_f + N_r)$) to perform unlearning. In contrast, other methods require $k$ iterative calculations of gradients, and careful tuning of hyper-parameters, such as learning rate, to achieve a balance between unlearning effectiveness and utility preservation. See Table 1, where if the learning rate is high (e.g., $10^{-6}$), utility is compromised; and if the learning rate is low (e.g., $10^{-7}$), unlearning is not effective.

**Localizing layers.** Our method efficiently identifies critical layers for updates, reducing the search space from hundreds to just a few. Figures 2, 7, and 12 show which layers are selected for unlearning different identities. This is achieved by combining two key

metrics: layer importance, which measures how sensitive the forget loss is to changes in each layer, and gradient alignment, ensuring updates minimally affect the retain set. These metrics help identify Pareto-optimal layers that balance effective unlearning with preserving model utility (explained further in Section 3). Our approach also reveals the distinct roles of layers in different architectures. Across various identities (see Figure 7) and architectures (see Figure 12), final projection layers of vision and language models are often updated due to their role in transforming complex features into final predictions. We also observe that the late attention layers in vision models and early attention layers in language models are selected for updates. Vision transformers utilize attention mechanisms to focus on different parts of an image and aggregate contextual information from various spatial regions. The late attention layers in these models are closer to the output; thus, more specialized in refining and utilizing contextually rich, high-level features. In contrast, language models often employ attention mechanisms right from the early layers to capture and process the sequential and contextual dependencies inherent in the textual data. Early attention layers are crucial for establishing a foundational understanding of the language structure, including syntax and semantics. By focusing on early layers, modifications can influence the foundational processing of input text, effectively guiding the subsequent layers' interpretation and response to the content.

### 4.3 UNLEARNING FOR STABLE DIFFUSION

Stable Diffusion (SD) models exhibit remarkable capabilities in text comprehension and the generation and manipulation of personal images. For instance, when prompted with "A portrait of Elon Musk", SDs can produce a high-fidelity portrait. Moreover, by altering the prompt, one can generate imaginative content, such as a vivid depiction of "Elon Musk on Mars". However, the potential misuse of such powerful tools raises significant concerns regarding the harm they can cause to the data provider (Yang et al., 2024).

**Unlearning identity.** In this study, we demonstrate how to effectively erase personal information from the image generation model, ensuring that prompts related to the erased individual fail to produce accurate results. Figure 4 presents examples of images generated by SDs before and after unlearning. Our approach to unlearning Elon Musk interestingly results in representations of electronic circuits, consistently across various prompts, without compromising the model's general capability to generate a diverse range of other objects. In contrast, other methods not only struggle to accurately render portraits of other individuals but also degrade the image quality of unrelated objects. We provide additional results on unlearning more celebrity identities, and other case studies on unlearning copyright-protected content and novel concept, in Section H.

**Evaluation on UnlearnCanvas benchmark.** To further demonstrate the unlearning effectiveness and efficiency of SLUG, we also evaluate its performance on UnlearnCanvas (Zhang et al., 2024c), a benchmark focused on unlearning artistic style and object concepts in Stable Diffusion. It introduces a comprehensive set of metrics for both evaluating effectiveness and efficiency, including **UA** (Unlearn Accuracy), **IRA** (In-domain Retain Accuracy), and **CRA** (Cross-domain Retain Accuracy). The benchmark targets unlearning styles and objects from an SDv1.5 model fine-tuned to generate 20 different objects in 60 distinct styles. The benchmark utilizes target SD with the prompt: "A [object name] in [style name] style," to generate the unlearning dataset, comprising 20 images for each object-style pair (i.e., 400 images per style and 1,200 images per class), resulting in 24,000 images in total. We curate forgets set with images associated with each style/object for each unlearning objective.

In Table 2, we report the unlearning performance of SLUG in benchmark metrics, along with other state-of-the-art unlearning methods reported in UnlearnCanvas. Our method minimizes storage and computational time by only requiring the gradient values of a few layers on the Pareto front to be stored, and performing a one-step update along the gradient for unlearning. Despite being extremely efficient, our method does not suffer from significant performance degradation in any metric or task in UnlearnCanvas. Our method achieves excellent trade-off between unlearning and retaining accuracy. For qualitative evaluation, we provide visual examples in Section H.

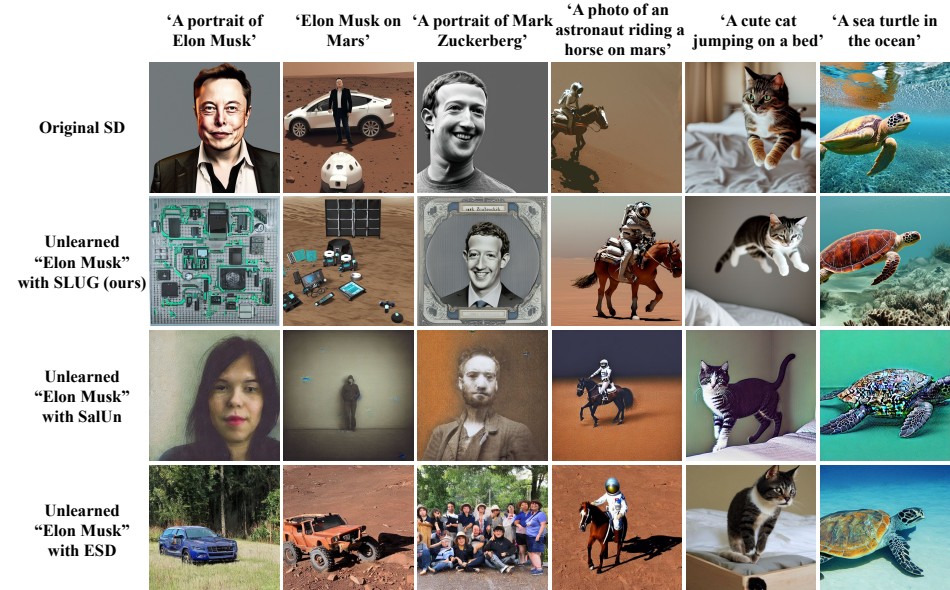

**Figure 4:** Images generated by different SDs using column captions as prompts. First row: images generated by the original pretrained SD. Second row: outputs of the SD after "Elon Musk" is unlearned using SLUG. We can see that "Elon Musk" is effectively unlearned, whereas other objects are unaffected. Bottom two rows: outputs of the SDs after "Elon Musk" is unlearned by existing methods (SalUn and ESD). We observe images generated for other unrelated prompts are also affected to some degree.

**Table 2:** Performance overview of different unlearning methods on UnlearnCanvas. The best performance for each metric is highlighted in green, and significantly underperforming results, in benchmark criteria, are marked in red. Our method SLUG shows no significant underperforming, and achieves the best trade-off among unlearning, retaining, and efficiency.

| Method | Effectiveness | | | | | | | Efficiency | | |
| | Style Unlearning | | | Object Unlearning | | | FID (↓) | Time | Memory | Storage |
| | UA (↑) | IRA (↑) | CRA (↑) | UA (↑) | IRA (↑) | CRA (↑) | | (s) (↓) | (GB) (↓) | (GB) (↓) |
|---|---|---|---|---|---|---|---|---|---|---|
| ESD (Gandikota et al., 2023) | 98.58% | 80.97% | 93.96% | 92.15% | 55.78% | 44.23% | 65.55 | 6163 | 17.8 | 4.3 |
| FMN (Zhang et al., 2024a) | 88.48% | 56.77% | 46.60% | 45.64% | 90.63% | 73.46% | 131.37 | 350 | 17.9 | 4.2 |
| UCE (Gandikota et al., 2024) | 98.40% | 60.22% | 47.71% | 94.31% | 39.35% | 34.67% | 182.01 | 434 | 5.1 | 1.7 |
| CA (Kumari et al., 2023) | 60.82% | 96.01% | 92.70% | 46.67% | 90.11% | 81.97% | 54.21 | 734 | 10.1 | 4.2 |
| SalUn (Fan et al., 2024) | 86.26% | 90.39% | 95.08% | 86.91% | 96.35% | 99.59% | 61.05 | 667 | 30.8 | 4.0 |
| SEOT (Li et al., 2024b) | 56.90% | 94.68% | 84.31% | 23.25% | 95.57% | 82.71% | 62.38 | 95 | 7.34 | 0.0 |
| SPM (Lyu et al., 2024) | 60.94% | 92.39% | 84.33% | 71.25% | 90.79% | 81.65% | 59.79 | 29700 | 6.9 | 0.0 |
| EDiff (Wu et al., 2024) | 92.42% | 73.91% | 98.93% | 86.67% | 94.03% | 48.48% | 81.42 | 1567 | 27.8 | 4.0 |
| SHS (Wu & Harandi, 2024) | 95.84% | 80.42% | 43.27% | 80.73% | 81.15% | 67.99% | 119.34 | 1223 | 31.2 | 4.0 |
| SLUG (Ours) | 86.29 ± 1.79% | 84.59 ± 1.63% | 88.43 ± 1.61% | 75.43 ± 2.91% | 77.50 ± 2.60% | 81.18 ± 1.46% | 75.97 | 39 | 3.61 | 0.04 |

## 4.4 Unlearning for VLMs

VLMs demonstrate impressive ability in visual understanding and question answering. For example, when provided with an image of a person, VLMs can accurately identify and name the individual depicted. Figure 5 demonstrates this by showing that when given an image of Elon Musk and asked, "What's the name of the person in this image?", the model correctly identifies him.

Our experiments focused on LLaVA-1.5, a popular VLM architecture. This model uses a pre-trained CLIP visual encoder to extract visual features from images. These visual features are then transformed into a format that can be understood by the language model. This transformation is done using a neural network layer that projects the visual information into the same space as word embeddings. The resulting visual tokens are then combined with language tokens and fed into the language model to generate responses. The key insight of our method is that the vision capability of VLMs heavily relies on the visual encoder. Therefore, by unlearning certain concepts from the CLIP vision model, we can influence the language model's understanding and generation of responses without directly modifying the language model itself. Figure 5 demonstrates the effectiveness of our approach. When given an image of Elon Musk and asked to identify the person, the original model correctly

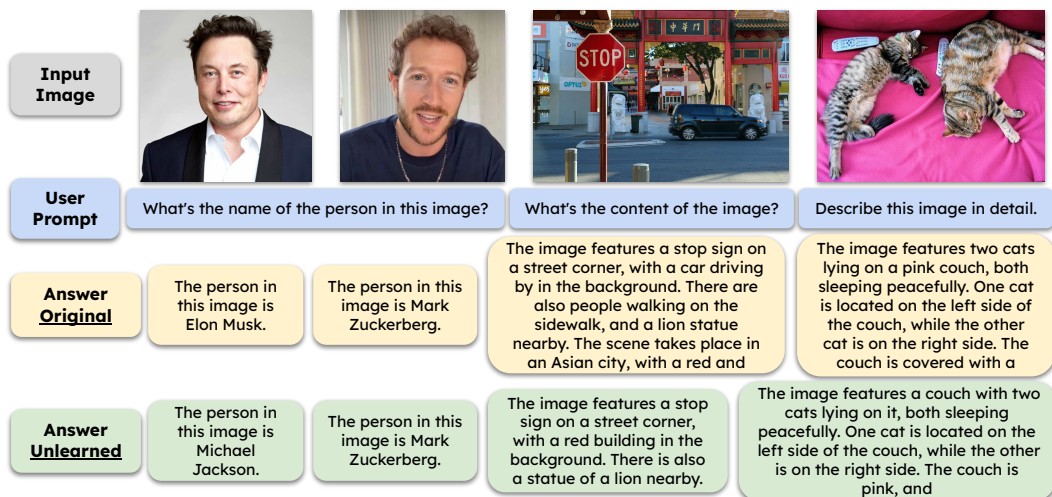

**Figure 5:** Effects of SLUG unlearning "Elon Musk" on LLaVA 1.5. The third row with yellow boxes shows the answers of the original model, and the forth row with green boxes shows the output of the unlearned model, where Elon Musk is effectively unlearned, whereas other concepts are unaffected.

names him. After applying our unlearning method, the model incorrectly identifies Elon Musk as Michael Jackson, indicating that the specific identity information has been successfully removed. This alteration does not significantly impact the model's overall utility. Additional examples of this phenomenon are discussed in Section I.

## 5 CONCLUSION

In this work, we introduced SLUG, an efficient machine unlearning method that requires just a single gradient computation and updates only one layer of the model. SLUG enhances unlearning feasibility on large models, especially with constrained hardware, while preserving overall model utility. Our experiments with CLIP, Stable Diffusion, and VLMs show that SLUG outperforms existing methods, particularly in unrelated tasks, with minimal computational overhead. SLUG's key innovation is its ability to identify and update only the most relevant layers, typically the late layers in vision models, when unlearning concepts like identities or copyrighted content.

This paper demonstrates that highly targeted, minimal interventions can be surprisingly effective for concept removal, suggesting that knowledge in neural networks may be more localized than previously thought. This has implications for our understanding of how information is encoded and stored in deep learning models. The ability to identify specific layers most relevant for particular concepts also provides new insights into the internal representations learned by different parts of neural networks. This contributes to the ongoing effort to improve the interpretability and transparency of AI systems.

**Limitations.** While SLUG shows clear advantages, there are limitations. Our experiments focused on vision-language models, and further testing is needed to evaluate its generalizability to other architectures, such as pure language models or graph neural networks. Additionally, we did not extensively explore long-term stability, adversarial resistance (Goel et al., 2022), or the ability to unlearn more abstract concepts. More work is needed to ensure robustness in adversarial scenarios and over extended periods or retraining.

## REPRODUCIBILITY STATEMENT

We are committed to ensuring reproducibility and have made our source code available, along with a comprehensive `README.md` to guide setup, execution, and replication of our results. Scripts and pre-computed gradients are also provided for easy reproduction of the main experiments in this paper.

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

## APPENDIX A   RELATED WORK

**Machine Unlearning** (Cao & Yang, 2015; Nguyen et al., 2022) has recently emerged as a critical area of research, driven by privacy concerns and regulatory requirements (gdp, 2016). Existing approaches mainly focus on a single task, like image classification (Liu et al., 2024b; neu, 2023; Guo et al., 2020; Goel et al., 2022; Chien et al., 2022; Golatkar et al., 2020b;a; Chundawat et al., 2023; Kurmanji et al., 2023; Jia et al., 2023; Shaik et al., 2023; Fan et al., 2024; Foster et al., 2024), image generation (Li et al., 2024a; Gandikota et al., 2023; Zhang et al., 2024a; Gandikota et al., 2024; Kumari et al., 2023; Li et al., 2024b; Lyu et al., 2024; Wu et al., 2024; Wu & Harandi, 2024), and LLMs text generation (Yao et al., 2024; Liu et al., 2024c). In this work, we propose a generic approach that is applicable to a wide range of multi-modal models including CLIP (Radford et al., 2021) for zero-shot image classification, stable diffusion models (Rombach et al., 2022) for text-to-image generation, and vision-language models (Liu et al., 2024a) for visual question answering.

For text-to-image diffusion models, particularly Stable Diffusion (SD), the evolution of unlearning approaches reveals increasing sophistication. Early methods such as ESD (Gandikota et al., 2023) and CA (Kumari et al., 2023) focused on modifying the UNet architecture through fine-tuning with negative guidance, but these approaches often resulted in widespread parameter updates across multiple layers, potentially compromising generation fidelity. More recent work has explored more targeted and efficient interventions. UCE (Gandikota et al., 2024) introduced a training-free unified approach using closed-form solutions for simultaneous debiasing, style erasure, and content moderation. FMN (Zhang et al., 2024a) achieved rapid concept removal through attention re-steering loss, redirecting generation from unwanted concepts to pretrained alternatives. SPM (Lyu et al., 2024) proposed an adapter-based approach using "concept-SemiPermeable Membranes" that can be flexibly transferred across different models without re-tuning. Other approaches include EDiff (Wu et al., 2024), which formulates unlearning as a constrained optimization problem to preserve model utility, and SEOT (Li et al., 2024b), which focuses on content suppression through text embedding manipulation and inference-time optimization. Despite these advances, existing methods still face challenges in balancing computational efficiency, generalization ability, and preservation of model utility, which our work aims to address through a principled single-layer approach.

**Saliency-based Methods.** Recent advances in machine unlearning have seen the emergence of saliency-based approaches, which aim to identify and modify only the most relevant parameters for concept removal. In image classification, methods like SSD (Foster et al., 2024) employ synaptic importance measures to selectively dampen connections, while SalUn (Fan et al., 2024) takes a simple and heuristic threshold-based approach. In text-to-image generation, SalUn (Fan et al., 2024) extend its framework by replacing cross-entropy loss in the unlearning objective to diffusion loss, requiring careful tuning of a gradient threshold for parameter selection. Diff-quickfix (Basu et al., 2024) utilizes causal inference with CLIPSscore (Hessel et al., 2021) as a metric to pinpoint concept-salient model parameters. MACE (Lu et al., 2024) proposes tuning the prompt-related projection matrices of the cross-attention blocks in the UNet architecture using LoRA modules (Hu et al., 2022). Similarly, CRE (Dong et al., 2024) identifies concept-specific causal denoising time steps in UNet layers and performs representation editing on selected layer outputs.

While these saliency-based methods represent the existing efforts in improving the efficiency of unlearning, their scope remains confined to specific tasks, such as image classification or text-to-image generation. Moreover, their parameter modifications often span multiple layers, which limits interpretability and flexibility in practical scenario. In contrast, our approach aims to extend efficient unlearning to foundation models that cover a diverse range of tasks (e.g., CLIP, Stable Diffusion, and vision-language models). By restricting model edits to a layer-specific scope, our framework introduces modularity to machine unlearning, abstracting the process into distinct layer updates along gradient vectors for tailored unlearning requests.

## APPENDIX B    ALGORITHM PSEUDO CODE

In this section, we present the pseudo code for our method, SLUG, in Algorithm 1, the search process for Pareto-optimal layers in Algorithm 2, and the binary search for the optimal unlearning step size in Algorithm 3.

Our implementation for the corresponding experimental models (i.e., CLIP, Stable Diffusion, and VLM) and benchmarks (i.e., UnlearnCanvas) has been made publicly available at https://anonymous.4open.science/r/SLUG-6CDF.

---

**Algorithm 1 SLUG**: Single Layer Unlearning Gradient

---

**Require:**
   Forget set $D_{\mathrm{f}}$ and retain set $D_{\mathrm{r}}$ ;
   Original model $F_{\theta}$ with model weights $\theta$;
   The set of all layers in the model, as $L$;
   Forget loss function $\mathcal{L}_{\mathrm{forget}}$ and retain loss function $\mathcal{L}_{\mathrm{retain}}$;
   Evaluation metrics forget accuracy FA and test accuracy TA.
**Ensure:** Unlearned model parameters $\theta_{\mathrm{f}}$
   1: Calculate and store $\nabla_{\theta}\mathcal{L}_{\mathrm{forget}}(\theta, D_{\mathrm{f}}), \nabla_{\theta}\mathcal{L}_{\mathrm{retain}}(\theta, D_{\mathrm{r}})$      ▷ *Single gradient calculation*
   2: **for** each layer $l$ in $L$ **do**
   3:    Importance$(l) = \|\nabla_{\theta_l}\mathcal{L}_{\mathrm{forget}}(\theta, D_{\mathrm{f}})\|_2 / \|\theta_l\|_2$      ▷ *Calculate layer importance*
   4:    Alignment$(l) = \cos(\nabla_{\theta_l}\mathcal{L}_{\mathrm{forget}}(\theta, D_{\mathrm{f}}), \nabla_{\theta_l}\mathcal{L}_{\mathrm{retain}}(\theta, D_{\mathrm{r}}))$      ▷ *Calculate layer alignment*
   5: **end for**
   6: $P = \mathbf{PO}(L, \mathrm{Importance}, \mathrm{Alignment})$      ▷ *Pareto optimal algorithm 2*
   7: $Q \leftarrow \varnothing$      ▷ *Set of layers and their performances*
   8: **for** each layer $l$ in $P$ **do**
   9:    $\lambda_0 = \mathrm{Importance}(l)/10$      ▷ *Initialize step size*
   10:    $(\lambda, \mathrm{FA}, \mathrm{TA}) = \mathbf{BS}(\lambda_0, l)$      ▷ *Binary search algorithm 3*
   11:    $Q \leftarrow Q \cup \{(l, \lambda, \mathrm{FA}, \mathrm{TA})\}$
   12: **end for**
   13: $\mathrm{FA}_{\mathrm{min}} = \min_{(l,\lambda,\mathrm{FA},\mathrm{TA}) \in Q} \mathrm{FA}$      ▷ *Identify minimum FA*
   14: $Q_{\mathrm{min}} = \{(l, \lambda, \mathrm{FA}, \mathrm{TA}) \in Q \,|\, \mathrm{FA} = \mathrm{FA}_{\mathrm{min}}\}$      ▷ *Filter sets with minimum FA*
   15: $(l^*, \lambda^*, \mathrm{FA}^*, \mathrm{TA}^*) = \arg\max_{(\lambda,\mathrm{FA},\mathrm{TA}) \in Q_{\mathrm{min}}}(\mathrm{TA})$      ▷ *Select set with highest TA*
   16: **return** $\theta_{\mathrm{f}} = \theta - \lambda^*\nabla_{\theta_{l*}}\mathcal{L}_{\mathrm{forget}}(\theta, D_{\mathrm{f}})$

---

**Algorithm 2 Pareto Optimal:** $P = \mathbf{PO}(L, \mathrm{Importance}, \mathrm{Alignment})$

---

**Require:**
   The set of all layers in the model, as $L$;
   Layer importance and gradient alignment of all layers
**Ensure:** The set of Pareto optimal layers
   1: Initialize $P \leftarrow \varnothing$      ▷ *Set of layers on the Pareto front is empty*
   2: **for** each layer $l$ in $L$ **do**
   3:    ParetoDominant $\leftarrow$ **true**
   4:    **for** each layer $l'$ in $L \setminus l$ **do**
   5:       **if** (Importance$(l') >$ Importance$(l)$ **and** Alignment$(l') <$ Alignment$(l)$) **then**
   6:          ParetoDominant $\leftarrow$ **false**
   7:          **break**
   8:       **end if**
   9:    **end for**
   10:    **if** ParetoDominant **then**
   11:       $P \leftarrow P \cup \{l\}$      ▷ *Identified a Pareto optimal layer*
   12:    **end if**
   13: **end for**
   14: **return** $P$      ▷ *Return the set of Pareto optimal layers*

---

---

**Algorithm 3** Binary Search for Optimal Step Size: $(\lambda^*, \texttt{FA}^*, \texttt{TA}^*) = \textbf{BS}(\lambda_0, l)$

---

**Require:**
    Initial step size $\lambda_0$;
    Maximum number of steps $K$;
    Model parameters $\theta$;
    Forget gradient of layer l: $G_l = \nabla_\theta \mathcal{L}_{\text{forget}}(\theta, D_{\text{f}})$
**Ensure:** Optimal $\lambda^*$, forget accuracy $\texttt{FA}$, test accuracy $\texttt{TA}$
1:  $\lambda_{\text{low}} \leftarrow 0$
2:  $\lambda_{\text{high}} \leftarrow \infty$
3:  $\lambda \leftarrow \lambda_0$
4:  $k \leftarrow 0$
5:  Initialize $P \leftarrow \varnothing$                                             ▷ *Performance set*
6:  **while** $k < K$ **do**
7:     $\texttt{FA}, \texttt{TA} = \texttt{eval}(\theta - \lambda G_l)$
8:     $P \leftarrow P \cup \{(\lambda, \texttt{FA}, \texttt{TA})\}$                        ▷ *Store results*
9:     **if** $\texttt{FA} > 0$ **then**
10:       $\lambda_{\text{low}} \leftarrow \lambda$                     ▷ *Should increase step size to unlearn*
11:     **else**
12:       $\lambda_{\text{high}} \leftarrow \lambda$            ▷ *Should reduce step size to avoid over-unlearning*
13:     **end if**
14:     **if** $\lambda_{\text{high}} == \infty$ **then**
15:       $\lambda \leftarrow 2\lambda$
16:     **else**
17:       $\lambda \leftarrow (\lambda_{\text{low}} + \lambda_{\text{high}})/2$
18:     **end if**
19:     $k \leftarrow k + 1$
20: **end while**
21: $\texttt{FA}_{\min} = \min_{(\lambda, \texttt{FA}, \texttt{TA}) \in P} \texttt{FA}$                      ▷ *Identify minimum FA*
22: $P_{\min} = \{(\lambda, \texttt{FA}, \texttt{TA}) \in P \,|\, \texttt{FA} = \texttt{FA}_{\min}\}$       ▷ *Filter sets with minimum FA*
23: $(\lambda^*, \texttt{FA}^*, \texttt{TA}^*) = \arg\max_{(\lambda, \texttt{FA}, \texttt{TA}) \in P_{\min}}(\texttt{TA})$     ▷ *Select set with highest TA*
24: **return** $\lambda^*, \texttt{FA}^*, \texttt{TA}^*$        ▷ *Select the set with lowest FA which has the highest TA*

---

## APPENDIX C    MORE EXAMPLES ON UNLEARNING IDENTITIES

In addition to the experiment on unlearning "Elon Mask" identity in the CLIP model, as discussed in Sec. 4.2 of the main text, we performed similar experiment on a broader set of identities: {Kanye West, Barack Obama, Bruce Lee, Fan Bingbing, Lady Gaga}. These names were selected from the CelebA dataset to represent a diverse cross-section of ethnicities and genders. Our method effectively identified the crucial layers associated with each name. These layers can then be specifically targeted to efficiently unlearn the corresponding identity from the CLIP model.

Figure 6 demonstrates that our approach successfully removes the desired names from the CLIP model (i.e., image-text alignment or cosine similarity becomes extremely low) . Figure 7 illustrates the Pareto-front plots that are used to identify important layers selected by our method for unlearning different identities.

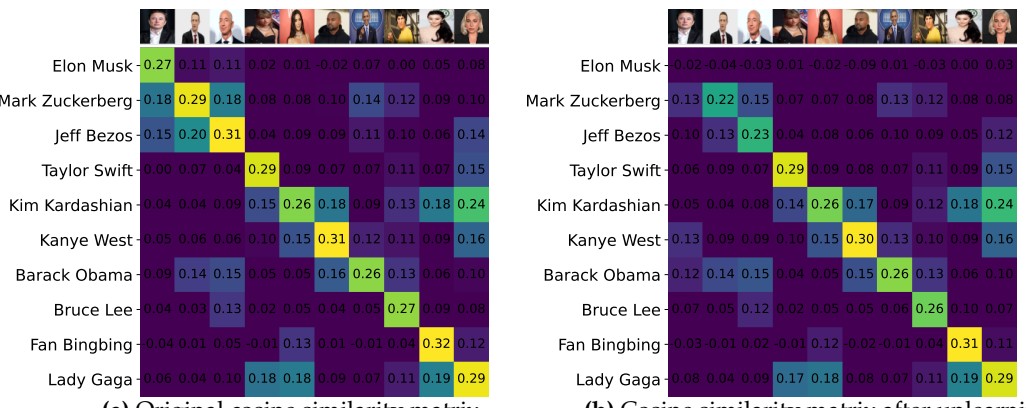

**(a)** Original cosine similarity matrix          **(b)** Cosine similarity matrix after unlearning

**Figure 6:** Cosine similarity matrix of image and text pairs before and after unlearning Elon Musk. After unlearning, the image and text pair of Elon Musk are not matched, while other persons are only slightly affected. Here the `vision attention out projection layer at the` $9_{th}$ `resblock` (associate with `9.attn.out_proj` in the pareto front legend) is unlearned. CLIP model: `ViT-B-16`

## APPENDIX D    JOINT UPDATE FOR UNLEARNING MULTIPLE IDENTITIES

We study the composite effect of our approach where we unlearn multiple tasks simultaneously. For instance, in the task of unlearning multiple identities, we use the gradients calculated for each identity on the original model and corresponding forget sets to identify the layers that are most significant for the respective identities, and then perform layer updates to simultaneously unlearn all of them. For joint updating, we follow the same updating scheme as described in Sec. 3. Firstly, different identities have different step size initialization from their respective gradients, and later on the step size is updated separately using binary search based on the unlearning result of the respective identity. We present our results in Figure. 8, where we successfully unlearn (a) {Elon Musk, Mark Zuckerberg} and (b) {Elon Musk, Taylor Swift}.

We also investigate how the unlearning performance varies as the number of identities to be forgotten increases. The identified layers are then updated in parallel to achieve unlearning of $N$ identities. Figure 9 demonstrate the effectiveness of our approach in unlearning $N$ identities for different values of $N$. Figure 7 presents details on identifying layers associated with different identities and updating them to achieve unlearning of multiple identities at once.

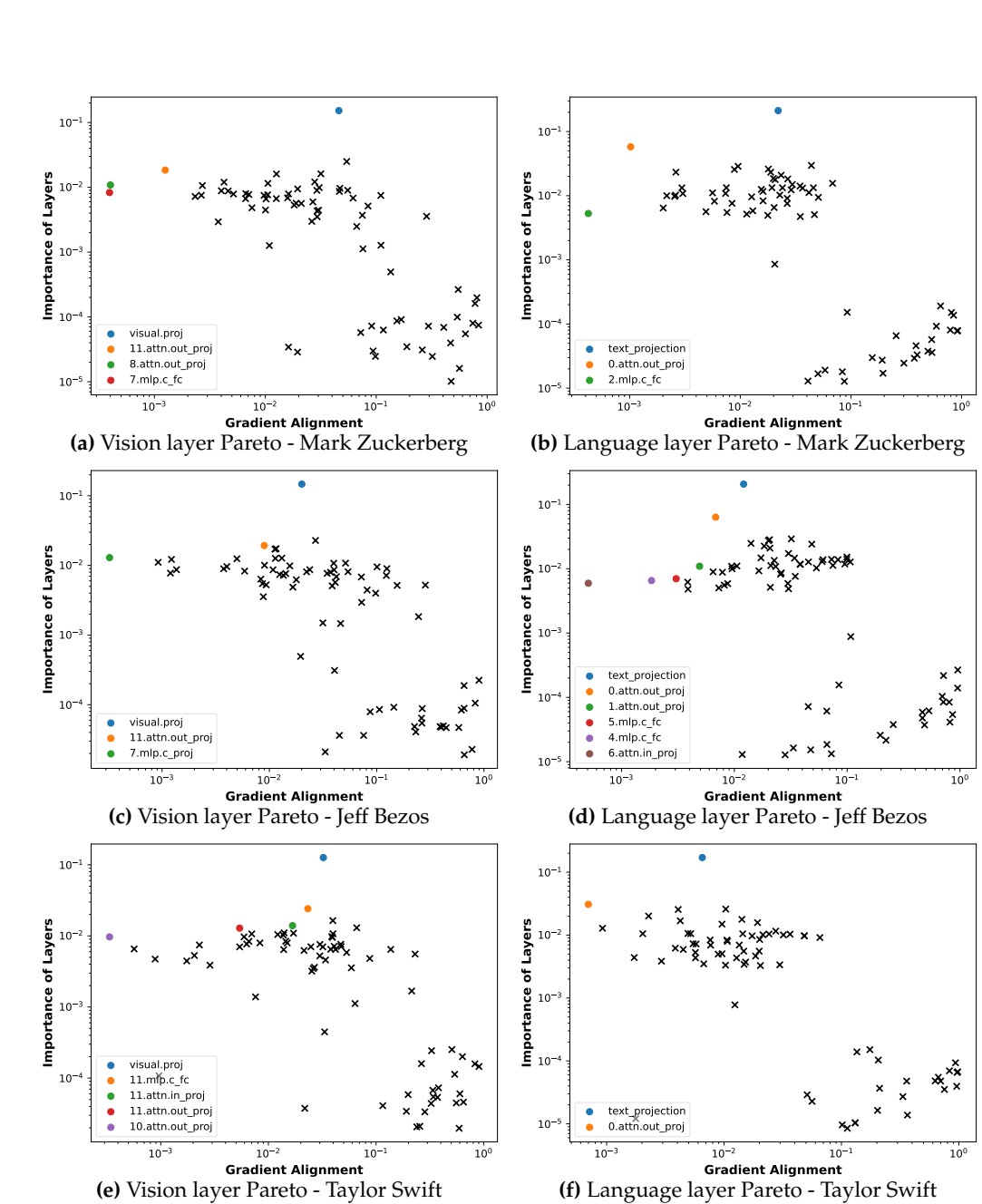

**(a)** Vision layer Pareto - Mark Zuckerberg

**(b)** Language layer Pareto - Mark Zuckerberg

**(c)** Vision layer Pareto - Jeff Bezos

**(d)** Language layer Pareto - Jeff Bezos

**(e)** Vision layer Pareto - Taylor Swift

**(f)** Language layer Pareto - Taylor Swift

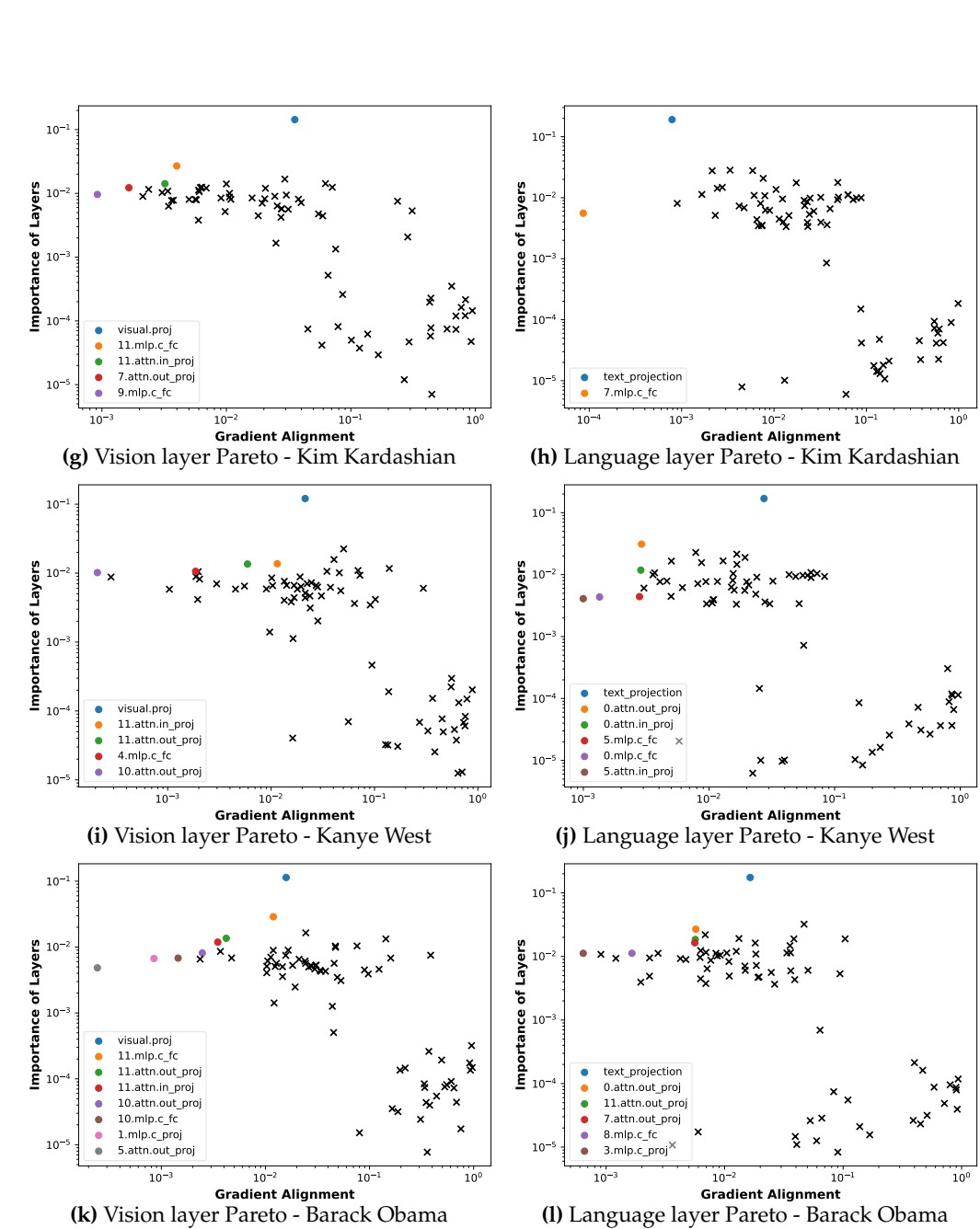

**(g)** Vision layer Pareto - Kim Kardashian   **(h)** Language layer Pareto - Kim Kardashian

**(i)** Vision layer Pareto - Kanye West   **(j)** Language layer Pareto - Kanye West

**(k)** Vision layer Pareto - Barack Obama   **(l)** Language layer Pareto - Barack Obama

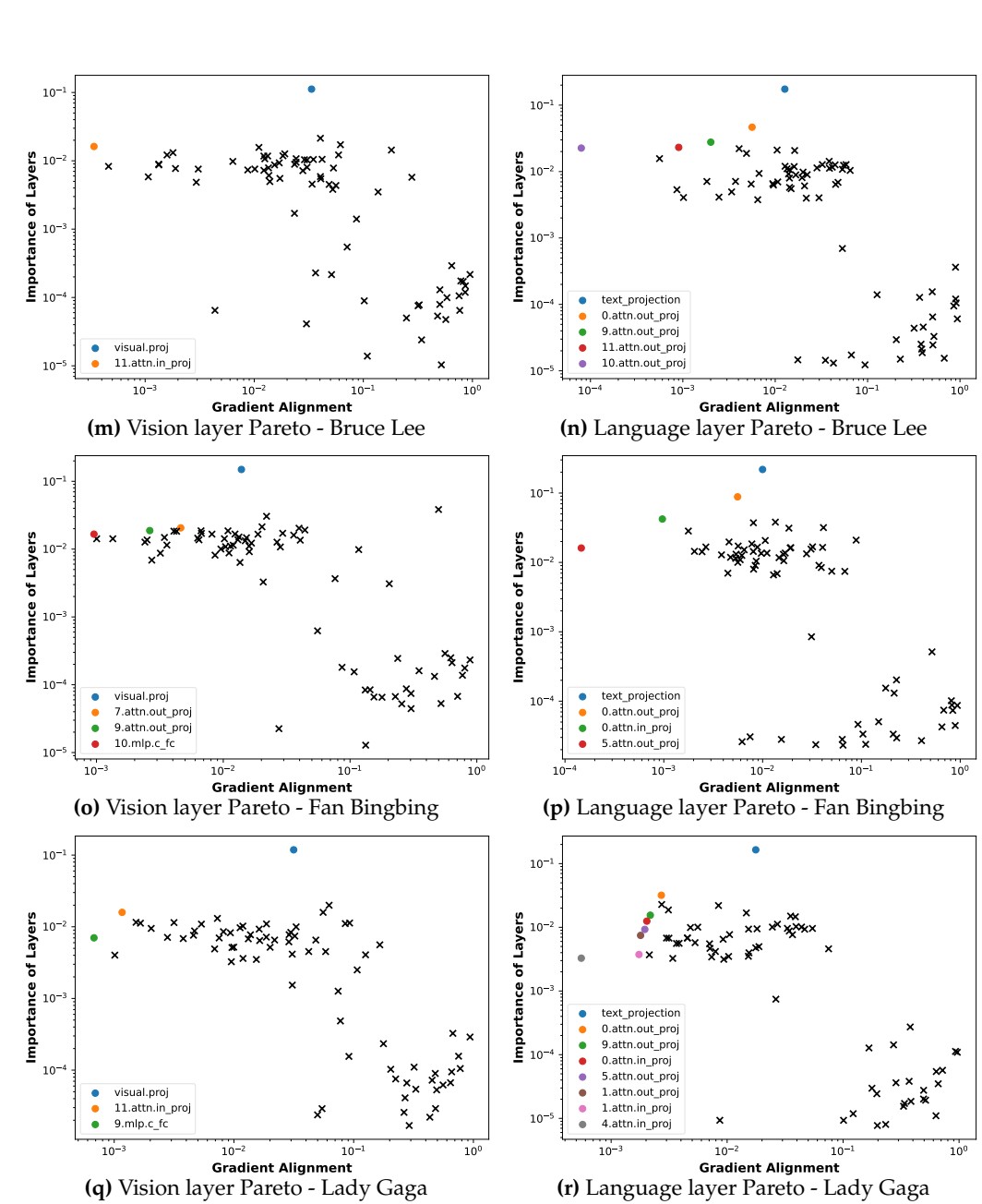

**Figure 7:** Scatter plots of layers for unlearning more identities, same setting as Figure 2. CLIP model `ViT-B-32`. Figures (a) - (r) shows the importance and gradient alignment of different vision model and language model layers as we unlearn different identities.

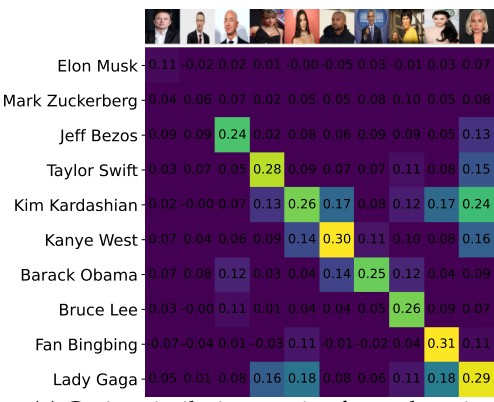
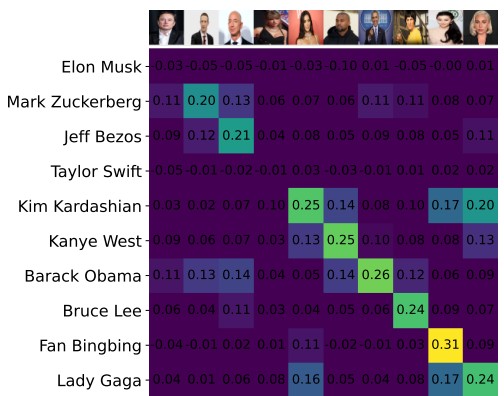

**(a)** Cosine similarity matrix after unlearning Elon Musk and Mark Zuckerberg

**(b)** Cosine similarity matrix after unlearning Elon Musk and Taylor Swift

**Figure 8:** Cosine similarity matrix of image and text pairs after unlearning multiple name identities (see Figure 6a for cosine similarity matrix on original model). (a) Unlearning Elon Musk and Mark Zuckerberg. (b) Unlearning Elon Musk and Tylor Swift. In both cases, the image and text pair of selected identities are not matched after unlearning, while other identifies are only slightly affected. We selected and updated the vision layer `9.attn.out_proj` for Elon Musk and the vision layer `11.attn.out_proj` for the other identity according to the pareto fronts in Fig. 7a and Fig. 7e, in both (a) and (b). We used CLIP model: `ViT-B-32` for these experiments.

# APPENDIX E   MORE CLIP MODELS

We performed experiments using an expanded set of model architectures. The results for {`ViT-B-16` are discussed above in Figure 6. The results for `ViT-L-14, EVA01-g-14`} are discussed in Figures 10,11, respectively. Figure 12 shows the metrics for different layers that our method uses to identify significant layers. These results demonstrate our method offers scalability and effectiveness across a range of model sizes, from 149.62 million parameters (`ViT-B-16`) to 1.136 billion parameters (`EVA01-g-14`). This underscores the flexibility of our approach to accommodate models of different scales.

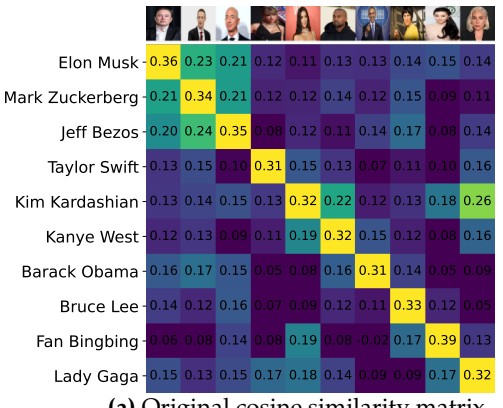
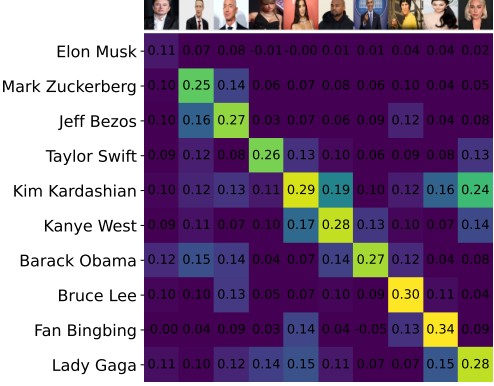

**(a)** Original cosine similarity matrix

**(b)** Cosine similarity matrix after unlearning

**Figure 10:** Cosine similarity matrix of image and text pairs before and after unlearning Elon Musk. After unlearning, the image and text pair of Elon Musk are not matched, while other persons are only slightly affected. Here, based on the pareto front in Fig. 12c, we select and update the vision layer `23.mlp.c_fc` for unlearning. CLIP model: `ViT-L-14`

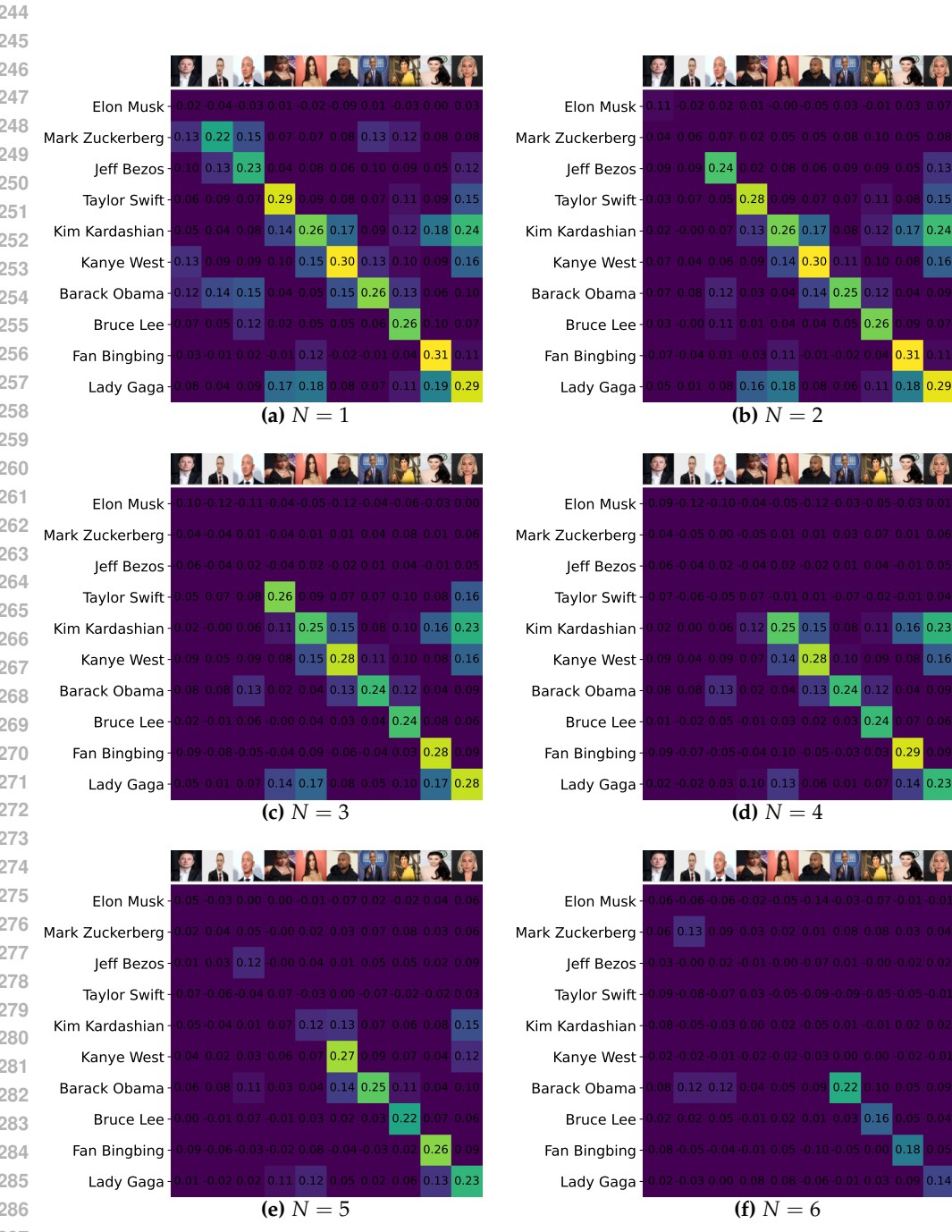

**Figure 9:** Cosine similarity matrices as we unlearn $N$ identities, where $N \in \{1, 2, ..., 6\}$. (a)–(f) Unlearn Elon Musk, Mark Zuckerberg, Jeff Bezos, Taylor Swift, Kim Kardashian, and Kanye West in a joint manner. To unlearn $N$ identities, our method (SLUG) identifies up to $N$ layers in the model using the single gradient calculated with the original network weights. The identified layers are then updated in parallel to achieve unlearning of $N$ identities.

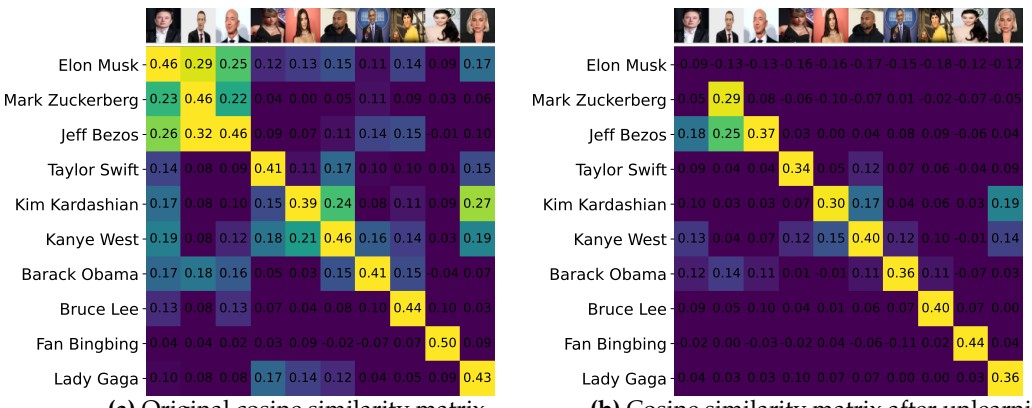

**(a)** Original cosine similarity matrix      **(b)** Cosine similarity matrix after unlearning

**Figure 11:** Cosine similarity matrix of image and text pairs before and after unlearning Elon Musk. After unlearning, the image and text pair of Elon Musk are not matched, while other persons are only affected. Here, based on the pareto front in Fig. 12f, we select and update the language layer `11.attn.out_proj` for unlearning. CLIP model: `EVA01-g-14`.

## APPENDIX F   UNLEARN DIFFERENT CONCEPTS

In addition to unlearning identities from CLIP, we also sample 7 classes {Basketball, Beach, Castle, Revolver, Rifle, School bus, Sunglasses} from ImageNet to evaluate the unlearning performance of our method on object concepts. For this experiment, we use 10k ImageNet validation images and sample images associated with target classes to create forget sets and compute gradients to unlearning different classes from the CLIP model. For evaluation, we use zero-shot accuracy reduction as the metric of effective unlearning target classes from the CLIP. The results, presented in Table. 3, show the CLIP zero-shot accuracy evaluations for both the forgetting of sampled classes and the retention of other ImageNet classes after unlearning. Our findings indicate that our method effectively reduces the CLIP zero-shot accuracy for the targeted classes to 0.0%, while the accuracy for remaining classes remains high, experiencing only minimal degradation (ranging from 0.03% to 2.03%) compared to the original pre-trained model, which indicates that the model's original functions are highly preserved after our unlearning.

**Table 3:** Unlearning performance of our method on common object concepts. FA@1 and FA@5 represents the top-1 and top-5 forget accuracy (%) of each forget class (i.e., zero-shot classification accuracy of unlearned class). TA@1 and TA@5 represents the top-1 and top-5 accuracy (%) of all classes of ImageNet except the corresponding Forget class. Each row shows the forget class accuracy and average accuracy over all classes of ImageNet before and after unlearning a class. Our method can reduce the forget accuracy of Forget classes to 0.0% while keeping the accuracy of the remaining classes close to original model (within $0.06 - 2.03\%$ difference). CLIP model: `ViT-B-32`. TA@1 and TA@5 for the original model remains almost the same for all rows; therefore, we list it once in the table.

| Forget class | Original | | | | Unlearned | | | |
| --- | --- | --- | --- | --- | --- | --- | --- | --- |
| | FA@1 | FA@5 | TA@1 | TA@5 | FA@1 ↓ | FA@5 ↓ | TA@1 ↑ | TA@5 ↑ |
| Basketball | 100.0 | 100.0 | | | 0.0 | 0.0 | 59.18 | 84.48 |
| Beach | 54.55 | 72.73 | | | 0.0 | 0.0 | 59.54 | 84.78 |
| Castle | 87.50 | 100.0 | | | 0.0 | 0.0 | 58.13 | 83.87 |
| Revolver | 100.0 | 100.0 | 60.16 | 85.52 | 0.0 | 0.0 | 59.94 | 85.43 |
| Rifle | 42.86 | 57.14 | | | 0.0 | 0.0 | 60.08 | 85.49 |
| School bus | 76.92 | 100.0 | | | 0.0 | 0.0 | 59.50 | 89.18 |
| Sunglasses | 44.44 | 55.56 | | | 0.0 | 0.0 | 60.13 | 85.23 |

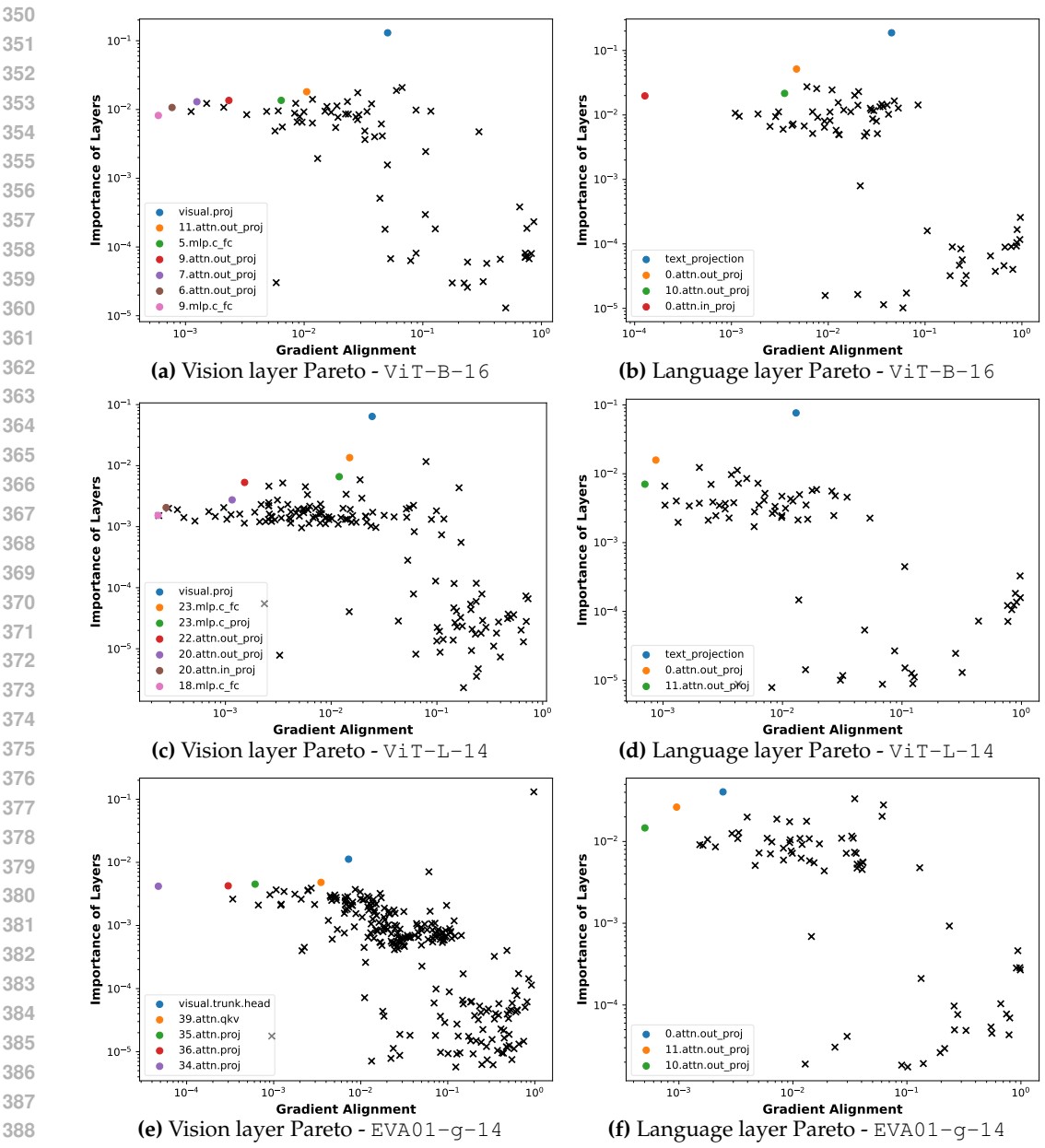

**(a)** Vision layer Pareto - `ViT-B-16`

**(b)** Language layer Pareto - `ViT-B-16`

**(c)** Vision layer Pareto - `ViT-L-14`

**(d)** Language layer Pareto - `ViT-L-14`

**(e)** Vision layer Pareto - `EVA01-g-14`

**(f)** Language layer Pareto - `EVA01-g-14`

**Figure 12:** More CLIP models, in addition to Sec 4.2. Unlearning name `Elon Musk` from different CLIP models built in: {`ViT-B-16`, `ViT-L-14`, and `EVA01-g-14`}

# APPENDIX G  LINEARITY OF UNLEARNING TRAJECTORY OF DIFFERENT LAYERS

In addition to the layers presented in Figure 2 (c) and (d), we show in Figure 13 that different layers show similar unlearning behaviors if we update them along their respective gradient direction (computed once for the original model). Nevertheless, the utility performance may vary depending on the selected layer; thus, it is important to select the best layer from the Pareto set for the overall best performance.

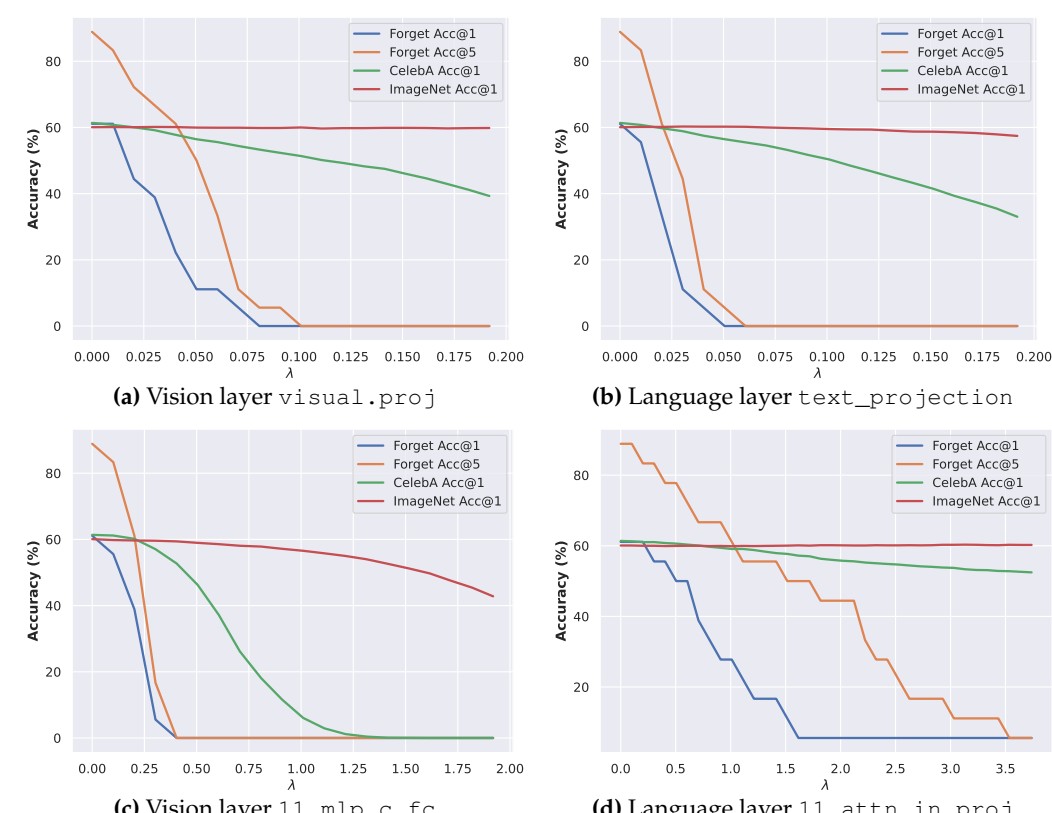

**(a)** Vision layer `visual.proj`

**(b)** Language layer `text_projection`

**(c)** Vision layer `11.mlp.c_fc`

**(d)** Language layer `11.attn.in_proj`

**Figure 13:** More examples of unlearning different layers. Correspond to Figure 2. The performance changes monotonically with the step size $\lambda$.

## APPENDIX H    MORE EXAMPLES ON STABLE DIFFUSION

To demonstrate the performance and practical utility of our method, we further consider unlearning more celebrity names and more scenarios including unlearning copyright characters, novel concepts and artistic styles on Stable Diffusion.

**More celebrity names.** Beyond unlearning "Elon Musk" from Stable Diffusion, which is presented in the Figure 4, here we also provide additional qualitative evaluations on unlearning other celebrity names {`Taylor Swift, Jeff Bezos`} with our method in Figure 14.

**Unlearning concepts and copyright content.** In addition to identity removal for privacy protection, we address copyright concerns that increasingly challenge generative models. For unlearning copyrighted contents from Stable Diffusion models, we generate 500 images using unlearning targets as prompts, and use them as the forget set. The retain set is a single shard of LAION-400M dataset, same as for CLIP unlearning.

We successfully apply our method to remove copyright-protected content, specifically targeting well-known characters such as Marvel's "Iron Man" and Walt Disney's "Mickey Mouse." Figure 15 illustrates that our technique precisely unlearns the targeted concepts, effectively disabling the generation of images associated with these copyrighted entities while preserving the ability of the model to produce images of other concepts. These results demonstrate the use of SLUG in protecting intellectual property from generative AI.

**Novel concept.** One of the intriguing properties of the Stable Diffusion is its ability to generalize image generation to novel concepts that are infrequently or never observed in the real world. In this experiment, we explore the unlearning of a unique concept, "Avocado chair" from Stable Diffusion. We first generate 500 image using SD with the prompt "An

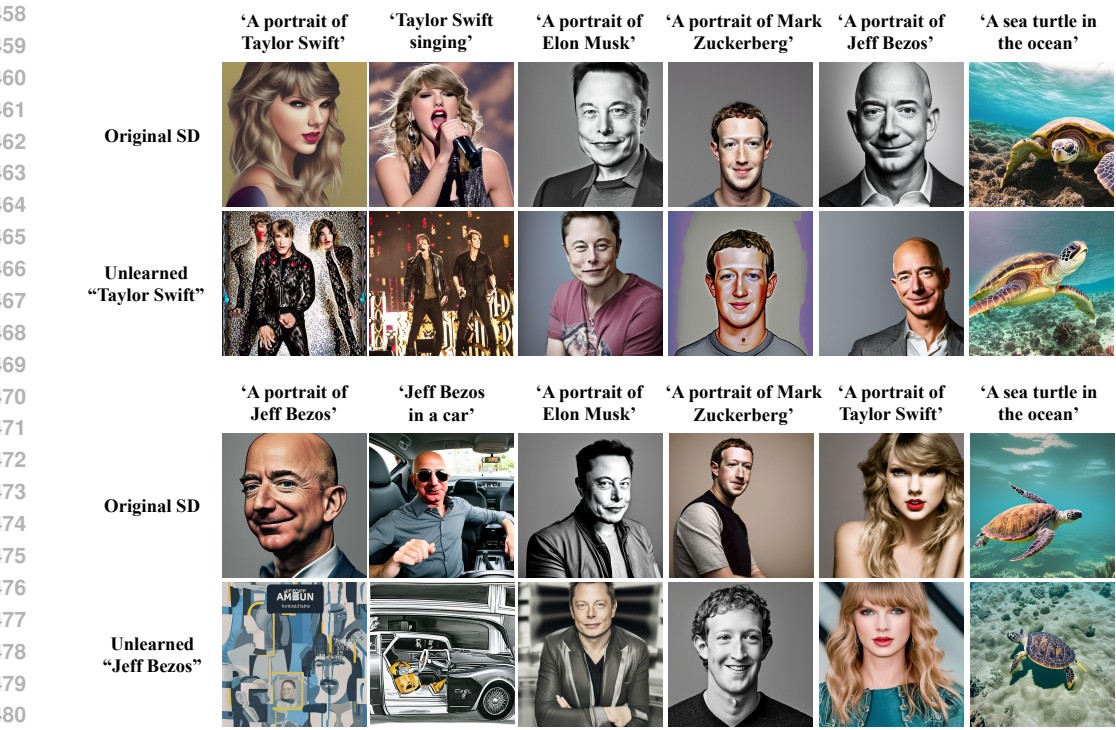

**Figure 14:** Qualitative evaluation on unlearning celebrity names Taylor Swift and Jeff Bezos from the Stable Diffusion.

avocado chair" to create the forget set, and use the same retain set as other experiments, which is is a single shard of LAION-400M dataset. In Figure 16, we show that our method successfully unlearn the concept "Avocado chair" from SD, resulting in the model's inability to generate images corresponding to this specific concept.

It is noteworthy that the model's capability to generate images related to the constituent atomic concepts (namely "Avocado" and "Chair") is also compromised. We hypothesize that this occurs due to the model's treatment of novel concepts as compositions of atomic concepts. For example, the concept "Avocado chair" is interpreted by the model as "Avocado" plus "Chair." Consequently, when a novel concept is unlearned, the associated atomic concepts are inadvertently affected as well. This highlights a challenge in the model's approach to handling the interoperability of novel and atomic concepts.

**Artistic styles and object.** In the experiment of evaluating SLUG performance on Unlearn-Canvas benchmark discussed in Section. 4.3, we use 400 images that are associated with each style, as the forget set for unlearning style, and 1200 images that are associated with each object concept as the forget set for unlearning object, all images are from the benchmark dataset. We use a single shard of LAION-400M dataset as the retain set.

For qualitative evaluation of this experiment, we provide visual examples of unlearning artistic styles: {Pop Art, Crayon, Sketch, Van Gogh} and object: dog that are sampled from UnlearnCanvas, in Figure 18, 19 and 20. These results further show the effectiveness of SLUG in unlearning a broad spectrum of concepts ranging from concrete (e.g., celebrity name, intellectual property figure, and object) to abstract (e.g., novel concept and artistic style).

APPENDIX I    MORE EVALUATIONS ON VLM

In addition to results presented in the main text Figure 5, we also present additional qualitative results on unlearning a different name "Taylor Swift" from VLM in Figure 17. We demonstrate that our method can anonymize celebrity names from the pretrained Vision-

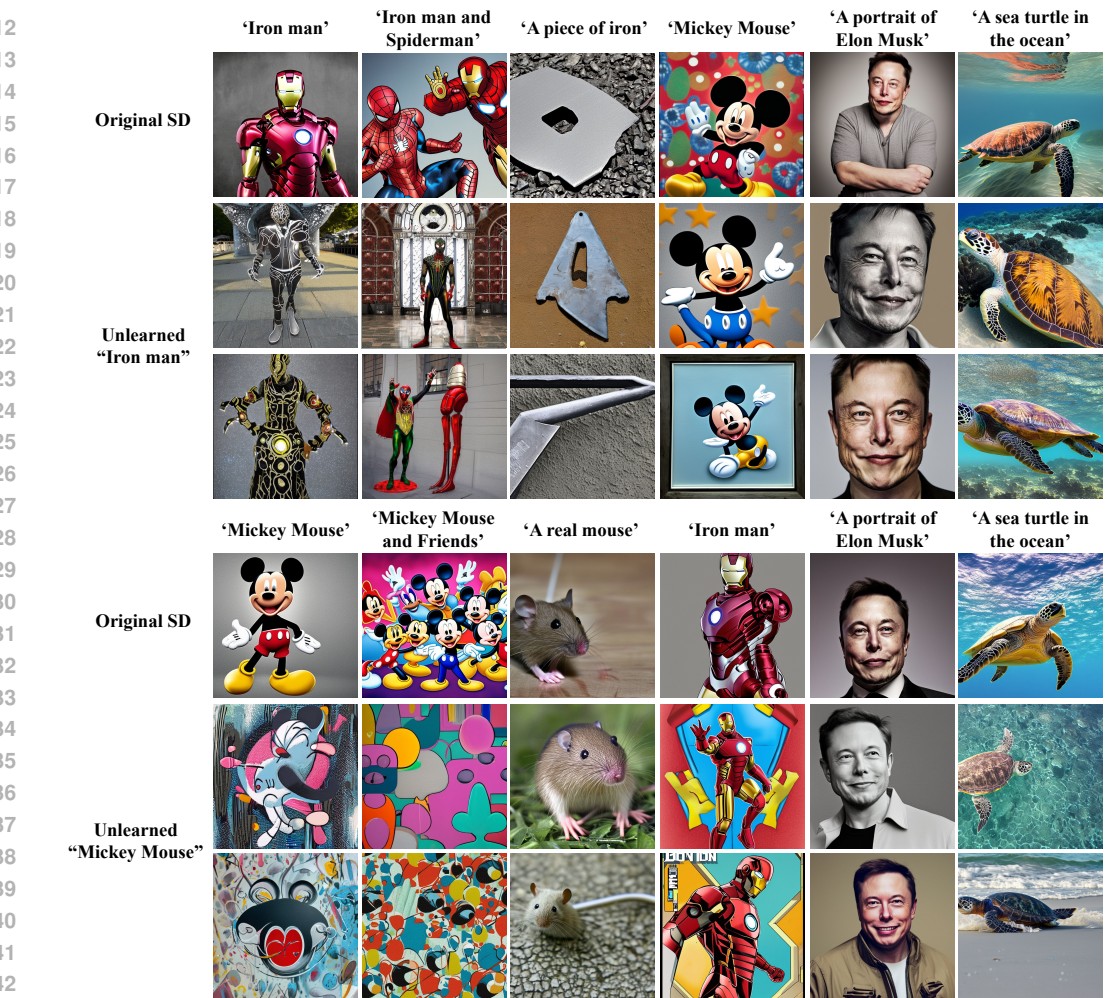

**Figure 15:** Qualitative evaluation on unlearning copyright characters "Iron man" and "Mickey Mouse" from SD, in first and second groups of figures respectively. First row shows the generated images from the original pretrained model, the second and third rows show the output of unlearned model using prompts captioned at the top of each column. Our method precisely unlearned copyright protected concepts from SD, while the image generation quality on other concepts is highly preserved.

**Table 4:** Quantitative evaluation on unlearning LLaVA-1.5.

| Model | FA ($\downarrow$) | VLM Benchmark Score ($\uparrow$) | | | |
|---|---|---|---|---|---|
| | | MME Cognition | MME Perception | GQA | MMBench (en) |
| Original LLaVA-1.5 | 99.50 | 323.57 | 1481.21 | 61.28 | 62.97 |
| Unlearned "Elon Musk" | 3.0 | 298.57 | 1354.61 | 60.70 | 61.34 |
| Unlearned "Taylor Swift" | 2.0 | 334.64 | 1336.09 | 60.72 | 60.14 |
| Average | 2.5 | 316.61 | 1345.35 | 60.71 | 60.74 |

language models, and simultaneously preserve the model's ability on image understanding, reasoning and distribution shift robustness on art work, cartoon style images.

We perform additional experiments for quantitative evaluations of the VLM model (LLaVA-v1.5-7B) that we qualitatively analyzed in Figure 5 and 17. Specifically, we evaluate two instances of LLaVa-v1.5 unlearned for two targeted identities (Elon Musk and Taylor Swift)

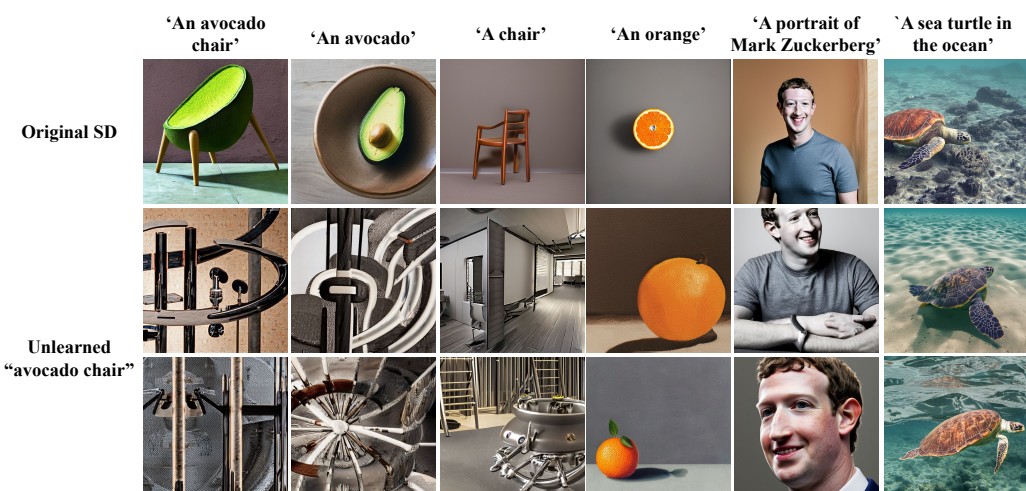

**Figure 16:** Qualitative evaluation on unlearning a novel concept "Avocado chair" from the SD.

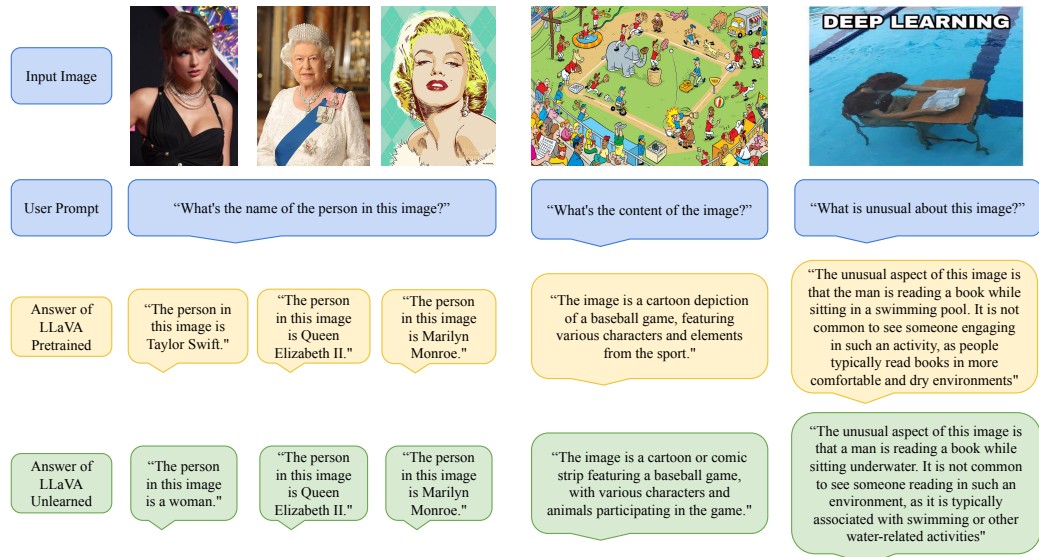

**Figure 17:** Qualitative evaluation on unlearning name "Taylor Swift" from LLaVA 1.5. While "Taylor Swift" is mapped to "woman" after the unlearning, the other female celebrity identification remain unaffected. Besides, model's robustness against style distribution shift is also preserved.

on three established VLM benchmarks: MME (Fu et al., 2023) GQA (Hudson & Manning, 2019), and MMBench (Liu et al., 2025). Higher benchmark scores indicate better performance of a VLM.

The results in Table 4 highlight that SLUG achieves effective unlearning while maintaining utility, validating its effectiveness in the VLM context. For forget accuracy, we tested each targeted celebrity using 100 images associated with that identity. The forget accuracy evaluation involves the question, "What is the name of the person in the image?" with the corresponding celebrity name as the correct answer. The benchmark scores represent the utility of the model for vision-language tasks, which contains a broad set of coarse-to-fine-grained questions on visual recognition and visual reasoning.

Overall, our results demonstrate that unlearned models accuracy on the targeted identity drops significantly, while benchmark scores remain high and close to those of the original model, preserving its overall utility.

## APPENDIX J    EXPERIMENT DETAILS ON UNLEARNCANVAS

**Models.** UnlearnCanvas targets unlearning styles and objects from an SDv1.5 model fine-tuned to generate 20 different objects in 60 distinct styles. The benchmark provides pre-trained SDv1.5 models for evaluation in `Diffusers` and `CompVis` implementations. In our experiment, correspondly, we focus on the CLIP text encoder used in SDv1.5 `Diffusers` implementation: `openai/clip-vit-large-patch14` from HuggingFace.

**Computational time, memory, and storage.** The gradient computational time and memory usage of SLUG depends on several factors: computing resource, batch size, and size of the forget set. Note that while the details of the evaluation of efficiency metrics are not well defined in the original UnlearnCanvas, in Table. 2 we are reporting the best performance of SLUG can achieve on our computing resource `NVIDIA A100 40GB`. Specifically, the batch size is set to 1 for recording the memory usage of SLUG, and to 16 for recording its computational time. This batch size of 16, is consistent with the sizes used in our other experiments.

For SLUG storage consumption, as our method only requires storing the gradient values of a few layers on the Pareto front, the actual storage consumption is 43 MB (0.043 GB), which by approximation is 0.0 GB in the original benchmark scale.

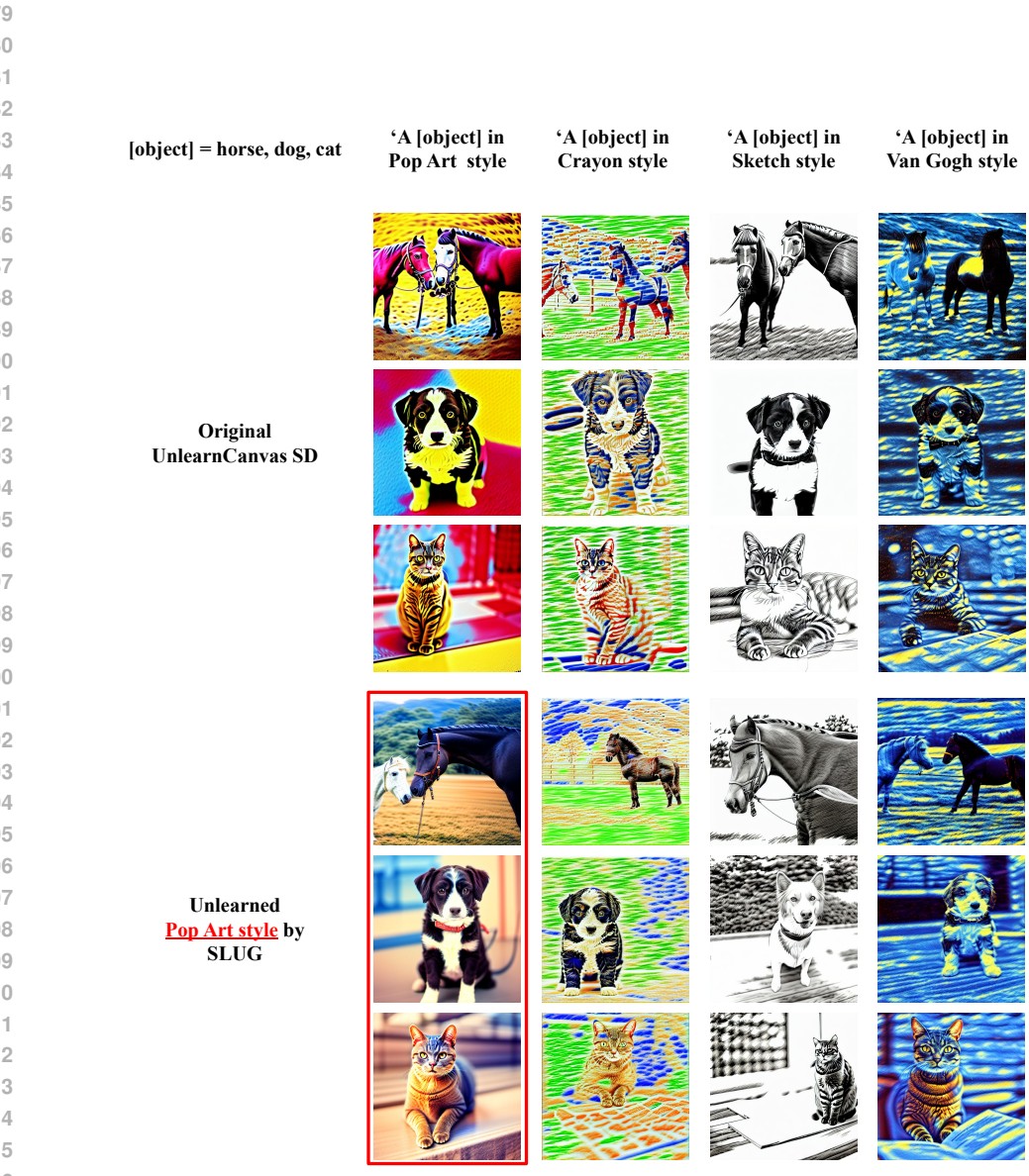

| [object] = horse, dog, cat | 'A [object] in Pop Art style | 'A [object] in Crayon style | 'A [object] in Sketch style | 'A [object] in Van Gogh style |

**Original UnlearnCanvas SD**

**Unlearned Pop Art style by SLUG**

**Figure 18:** Visual examples of SLUG performance on UnlearnCanvas. Row $1 - 3$: outputs from original UnlearnCanvas Stable Diffusion (SD) using column captions as prompts. Row $4 - 6$: outputs from UnlearnCanvas SD unlearned Pop Art style. Outputs corresponding to the unlearned style are highlighted by the red bounding box .

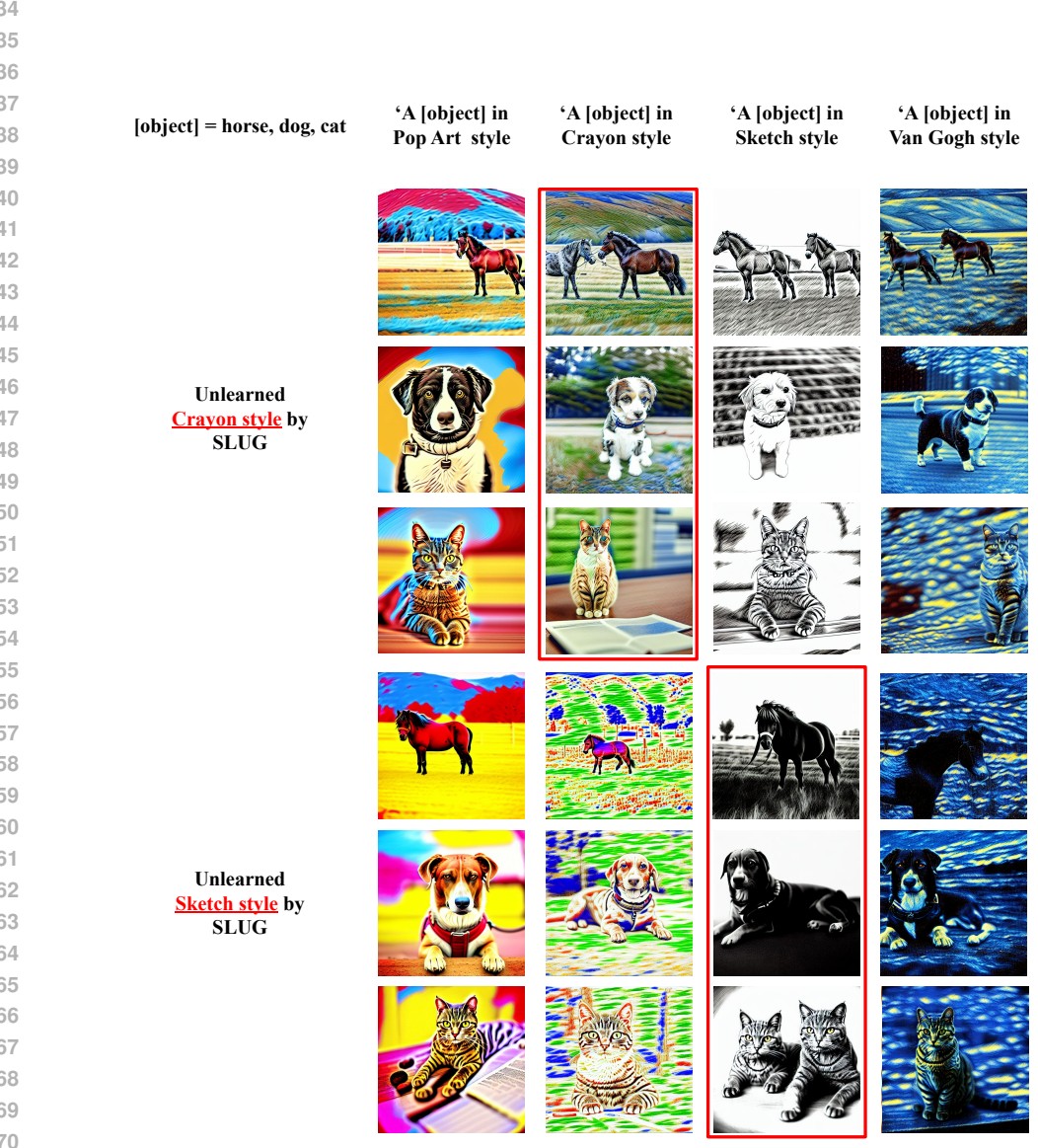

**Figure 19:** Visual examples of SLUG performance on UnlearnCanvas. Row $1 - 3$: outputs from UnlearnCanvas SD unlearned Crayon style. Row $4 - 6$: outputs from UnlearnCanvas SD unlearned Sketch style. Outputs corresponding to the unlearned style are highlighted by the red bounding box.

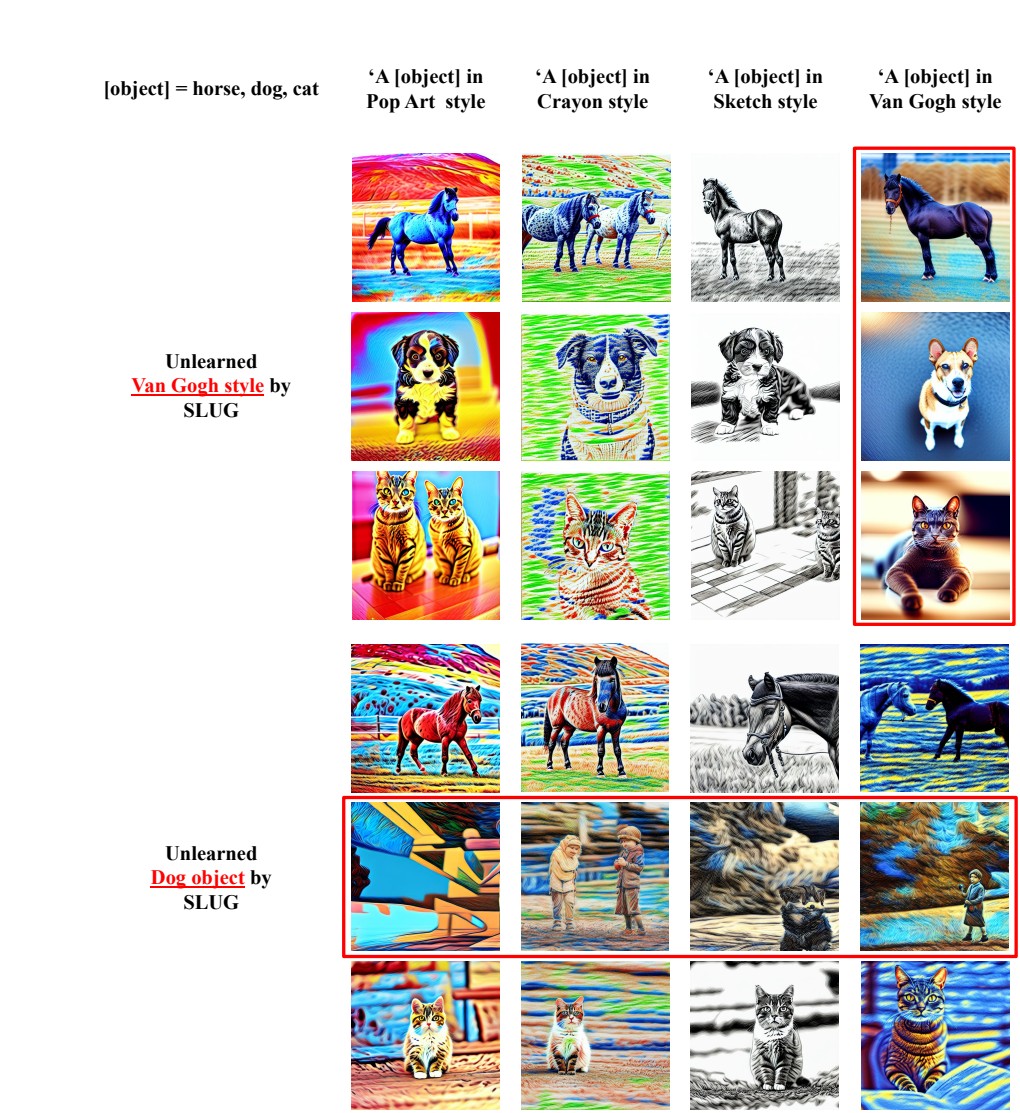

|  | 'A [object] in Pop Art style | 'A [object] in Crayon style | 'A [object] in Sketch style | 'A [object] in Van Gogh style |
|---|---|---|---|---|

[object] = horse, dog, cat

Unlearned **Van Gogh style** by SLUG

Unlearned **Dog object** by SLUG

**Figure 20:** Visual examples of SLUG performance on UnlearnCanvas. Row $1-3$: outputs from UnlearnCanvas SD unlearned Van Gogh style. Row $4-6$: outputs from UnlearnCanvas SD unlearned dog object. Outputs corresponding to the unlearned style/object are highlighted by the red bounding box .