# OpenReview forum: "Targeted Unlearning via Single Layer Unlearning Gradient"
_ICLR.cc/2025/Conference — Submitted to ICLR 2025_

### Official Review · Reviewer_AC4y · 2024-11-01

**Soundness:** 3
**Presentation:** 3
**Contribution:** 2
**Rating:** 5
**Confidence:** 2

**Summary:**

This paper introduces a method called Single Layer Unlearning Gradient (SLUG), aimed at addressing the challenges of unauthorized generation of privacy-related and copyright-infringing contents. SLUG is designed for unlearning of targeted information from trained machine learning models, requiring only a single gradient computation (and then reuse it) and updating only one layer of the model. This approach minimizes computational costs and maintains the model’s overall utility, particularly for unrelated tasks.
The method has been tested with popular models like CLIP and Stable Diffusion, demonstrating superior efficiency and effectiveness compared to existing methods.

**Strengths:**

## Strengths

1. **Balanced Unlearning and Performance:** The proposed method effectively balances the unlearning process with the model's general performance, addressing a crucial trade-off in model management.
2. **Computational Efficiency of SLUG:** SLUG requires gradient computation only once, offering two significant advantages:
   - **Faster Computation:** Reduces overall computation time.
   - **Prevention of Over-Unlearning:** Minimizes the risk of excessively removing learned information.
3. **Generalization Across Models:** SLUG demonstrates effectiveness not only on stable diffusion models but also yields promising results on Vision-Language Models (VLMs), showcasing its potential for broader applicability.

**Weaknesses:**

## Weaknesses

1. **Dependence on Retain Set:** SLUG relies on a retain set to preserve general performance. The methodology for curating this set is critical, yet the paper lacks sufficient discussion or guidelines to ensure reproducibility.
2. **Incomplete Computational Time Analysis:** While Table 1 presents a computation time comparison, the analysis based on $O(N_f + N_r)$ overlooks key factors:
   - **Iterative Parameter Updates:** SLUG requires iterative updates of model parameters as described in Equation 9.
   - **Layer Importance and Gradient Alignment:** The time associated with determining layer importance and performing gradient alignment is not accounted for, potentially underestimating the actual computational cost.
3. **Insufficient Evaluation on VLMs:** The claims regarding SLUG's performance on VLMs are not fully substantiated. More comprehensive experiments are necessary to convincingly demonstrate its superiority in this context.

**Questions:**

## Questions

1. **Retain Set Curation:** Could the authors provide a detailed explanation of how the retain set is curated? Clarifying this process is essential for reproducibility and assessing the method's robustness.
2. **Iterative Update Performance:** It is recommended to report the performance of the iterative update version of SLUG. If performance metrics decline, this could highlight underlying foundational issues that need to be addressed.

---

> ### Author Response · Authors · 2024-11-22
>
> Thank you for your detailed review and constructive feedback. We appreciate the thorough analysis and would like to address each of your main concerns (common concerns including dependence on the retain set, VLM experiments, and iterative update of SLUG are addressed above)
>
> **(Computational Time Analysis)** We use big O notation for gradient calculation complexity. For example, our method requires a one time calculation of gradients on the retain set (containing $N_r$ images) and the forget set ($N_f$ images); thus, we have the complexity of $O(N_f + N_r)$. For iterative gradient calculation like FT that takes k iterations using only forget set, the complexity is $O(k * N_r)$. Gradient calculation is the most memory and computation expensive operation. Other marginal computation costs such as calculating importance and gradient alignment metrics based on pre-calculated gradients and evaluating the models on the forget set are omitted for simplicity.
>
> In addition, we would like to clarify the iterative update. After gradient calculation, we only need to decide the correct step size for the gradient for which we perform a binary search on the gradient step size to find the boundary where the forget set performance is minimum. In this process, we need to run inference on the sampled forget set, which is a fast process and negligible compared to the gradient calculation. In Table 1, we evaluate the performance of other methods after each iteration to prevent over-unlearning, thus the omitted time is comparable across different methods.
>
> **(Q: Iterative update version of SLUG)**
> SLUG is an iterative unlearning scheme that updates a single layer along a single gradient direction. SLUG uses binary search for finding the suitable step size, which can be viewed as an iterative process.
>
> We report results for an “iterative SLUG” in Table R1 in the common comments above, where we performed an iterative gradient calculation on the single layer. Iterative gradient calculation and update of single layers achieves unlearning performance similar to single gradient direction update but slightly worse performance on CelebA classification (likely due to slight over-unlearning on identity). Iterative update also requires a careful learning rate tuning and early stopping; in contrast, single gradient direction offers an effective solution that is computationally simpler and robust to step size selection.
>
> We hope the comments above address your concern about the iterative update version of SLUG. If you need any other clarification, please let us know.
>
> Please let us know if you have any remaining questions, and we would be happy to address them.

---

> ### Author Response · Authors · 2024-12-03
> **We Hope Our Responses Address Your Concerns – A Friendly Reminder of the Discussion Deadline**
>
> Dear Reviewer AC4y,
>
> We greatly appreciate your time and effort in reviewing our work! In our earlier responses, we have carefully addressed your concerns by conducting the relevant experiments you highlighted and providing thorough explanations to clarify your questions about the paper.
>
> As the discussion period deadline approaches, we welcome any additional questions or feedback you may have. We would be delighted to continue the conversation and provide further clarification if needed.
>
> We look forward to hearing from you!
>
> Best,
> Authors

---

### Official Review · Reviewer_ufXX · 2024-11-03

**Soundness:** 3
**Presentation:** 3
**Contribution:** 3
**Rating:** 5
**Confidence:** 4

**Summary:**

The author proposes a method that only requires one-time gradient calculation to update a single layer of the model for achieving unlearning.  By approximating the importance of the measurement layer using the diagonal of the Fisher information matrix and balancing gradient alignment, the author selects a single target layer and finally updates its parameters in a single step to achieve the desired outcome. The effectiveness of this approach is validated through extensive experiments.

**Strengths:**

1. The paper is well-organized and easy to follow. This approach achieves effective unlearning with just a single gradient update on one layer, demonstrating remarkable efficiency, particularly in the context of large models.

2. In the proposed approach, the author employs the diagonal of the Fisher information matrix to approximate layer importance, thereby enhancing interpretability.

3. The author conducted extensive experiments on large-scale multimodal models including CLIP, Stable Diffusion, and VLMs, demonstrating the wide applicability of the proposed approach and empirically demonstrating its advantages in balancing efficiency and model utility. And the author provided complete code that supports reproducibility.

**Weaknesses:**

1. The proposed scheme only updates the most important layer to achieve excellent forgetting effects. Although the experimental results can provide an empirical guarantee for forgetting, intuitively there must be residual information in the remaining layers. From the experimental results, the difference in importance between layers is not large. Hence, it feels more reasonable to update as many layers as possible while maintaining model performance. It is better to add more discussions.
2. The design of the approach requires access to all forgotten and retained data. However, the targeted domain involves relatively large datasets, requiring substantial storage space. If complete access to the data is not feasible, could this negatively impact the effectiveness of the scheme?
3. The paper's description of layer selection is not clear enough, and I did not correspond the graph well with the Pareto optimal set. I cannot clearly understand how the author balances importance of layers and grade alignment.
4.  From the experimental results of unlearning for stable diffusion, it can be seen that unlearning leads to a slight decrease in the quality of image generation.
5. The experiment of unlearning for VLM lacks quantitative analysis and only shows examples. Adding quantitative analysis will provide clearer evidence for the method.

**Questions:**

Please check the questions in the weaknesses above.

---

> ### Author Response · Authors · 2024-11-22
>
> Thank you for your detailed review and constructive feedback. We appreciate the thorough analysis and would like to address each of your main concerns (common concerns including access to forget and retain data, layer selection, and VLM experiments are addressed above):
>
> **(Residual information in remaining layers)** While the intuition about residual information in other layers is reasonable, we argue that the success of unlearning should be evaluated by its functional impact rather than theoretical presence of information. Even for completely unseen data, neural networks contain transferable features across their layers (e.g., general facial features used for recognizing unseen faces). The key is whether this information can be effectively utilized for the targeted task. Our experimental results below in Table R4 actually demonstrate that updating multiple layers does not provide additional benefits. We compare the results of SLUG (update a single layer) with the results of updating all the layers on the pareto and in the network. We can observe that all approaches achieve complete forgetting (FA@1 = FA@5 = 0.00). However, SLUG maintains significantly better utility on related/desired tasks (TA_CA@1 = 58.32). Updating more layers actually degrades performance on related/desired tasks without improving forgetting.
>
> Table R4: Unlearning performance comparison between updating one layer and all layers (the experimental setup is consistent with Table 1 of the main paper).
>
> | Method               | FA@1 (↓)  | FA@5 (↓)  | TA_IN@1 (↑)  | TA_CA@1 (↑) |
> |----------------------|-----------|-----------|--------------|-------------|
> | All Pareto           | 0.00      | 0.00      | 59.92        | 51.64       |
> | All layers           | 0.00      | 0.00      | 59.70        | 53.74       |
> | SLUG (Ours, Table 1) | 0.00      | 0.00      | 59.96        | 58.32       |
>
> Our single-layer approach provides a better balance between targeted forgetting and preservation of general knowledge because it likely minimizes interference with layers that encode transferable features. SLUG focuses updates on the layer most directly responsible for the targeted concept and reduces the risk of catastrophic forgetting of related but desired concepts.
>
> **(Slight decrease in the image quality)** We agree that there is a slight decrease in the FID score for stable diffusion after unlearning, but we did not observe major change in the visual quality of generated images. In general, we observe a tradeoff between forgetting and utility preserving, and our method achieves a good balance of unlearning without significantly compromising the utility.
>
> Please let us know if you have any remaining questions, and we would be happy to address them.

---

> ### Author Response · Authors · 2024-12-03
> **We Hope Our Responses Address Your Concerns – A Friendly Reminder of the Discussion Deadline**
>
> Dear Reviewer ufXX,
>
> We greatly appreciate your time and effort in reviewing our work! In our earlier responses, we have carefully addressed your concerns by conducting the relevant experiments you highlighted and providing thorough explanations to clarify your questions about the paper.
>
> As the discussion period deadline approaches, we welcome any additional questions or feedback you may have. We would be delighted to continue the conversation and provide further clarification if needed.
>
> We look forward to hearing from you!
>
> Best,
> Authors

---

### Official Review · Reviewer_eMTT · 2024-11-04

**Soundness:** 3
**Presentation:** 2
**Contribution:** 2
**Rating:** 5
**Confidence:** 4

**Summary:**

This paper introduces a novel saliency-based method for the machine unlearning task. The proposed approach, named Single Layer Unlearning Gradient (SLUG), effectively removes targeted information by updating only a single specific layer of the model through a one-time gradient computation. Compared to traditional unlearning techniques, SLUG significantly reduces computational costs while ensuring minimal impact on the model's performance for unrelated content.

The authors evaluate SLUG using metrics such as low computational cost, effective unlearning, and utility retention. They demonstrate the method's efficacy across three downstream tasks: CLIP Zero-Shot Classification, Generative Models on UnlearnCanvas benchmark and Vision-Language Models.

**Strengths:**

1. **Innovative Saliency-Based Approach to Machine Unlearning**

   The paper introduces a novel saliency-based method specifically designed to address the machine unlearning problem. The authors present SLUG technique, which efficiently removes targeted information by updating only a single designated layer of the model through a one-time gradient computation. This method offers a streamlined solution compared to traditional unlearning techniques that often require extensive model modifications and incur high computational costs.

2. **Comprehensive Validation Across Diverse Downstream Tasks**

   The effectiveness of the proposed SLUG method is thoroughly validated across three distinct downstream tasks, demonstrating its versatility and robustness: CLIP-Based Image Classification, Stable Diffusion-Based Image Generation and Vision-Language Models (VLM).

**Weaknesses:**

1. **Lack of Related Work Discussion**

   The paper does not include a comprehensive review of related work. This omission makes it difficult to contextualize the proposed method within the existing body of research and to understand how it compares to or improves upon previous approaches in machine unlearning and saliency-based methods.

2. **Insufficient Clarity in Single Layer Update Methodology**

   The description of the **Single Layer Unlearning Gradient (SLUG)** method lacks clarity, particularly in the selection and updating of the single targeted layer. This can lead to confusion among readers regarding the following aspects:

   - **Balancing Equations (7) and (8)**: The paper does not adequately explain how these equations balance the unlearning process. Additional textual explanations are needed to clarify the interplay between these equations and their role in achieving effective unlearning.

   - **Computation of Single Gradient Direction**: The rationale behind choosing the gradient direction based on the initial parameters is not sufficiently elaborated. More detailed explanations are necessary to justify this choice and its impact on the unlearning process.

   - **Consistency in Parameter Updates**: Although the authors emphasize updating parameters in a single layer, this point is not clearly reiterated in Section 3.2. Ensuring consistent emphasis throughout the methodology section would enhance understanding.

3. **Limited and Inadequate Experimental Evaluation**

   The experimental results presented in the paper are not particularly compelling, and the evaluation metrics used are insufficiently comprehensive. Specific issues include:

   - **Unlearning for CLIP (Section 4.2)**:
     - **Optimal Results Visualization**: The results for different learning rates are not clearly highlighted. Using color-coding to indicate the best-performing results would improve readability and interpretation.
     - **Evaluation Metrics Consistency**: The paper does not maintain consistency with established definitions for classification unlearning tasks, such as those outlined in "Model Sparsity Can Simplify Machine Unlearning." Aligning the evaluation metrics with these definitions would strengthen the validity of the results.

   - **Unlearning for Stable Diffusion (Table 2)**:
     - **Limited Performance Advantages**: Beyond demonstrating efficiency, the method does not show significant advantages in other performance metrics. This limitation raises questions about the overall effectiveness of SLUG in this context.

   - **Application to Vision-Language Models (VLMs)**:
     - **Lack of Reported Data**: Although the paper highlights the application of the unlearning method to VLMs and mentions corresponding evaluation metrics, it fails to report the actual data results. This absence undermines the persuasiveness of the claims regarding the method's effectiveness in VLMs.

**Questions:**

Based on the weaknesses part, here are some corresponding suggestions:

1. **Incorporate a Comprehensive Related Work Section**

  If it is available, add a dedicated Related Work section that reviews pertinent literature on machine unlearning and saliency-based methods.

2. **Enhance Clarity in the Single Layer Update Methodology**

  The methodology for selecting and updating the single targeted layer is not clearly explained, potentially causing confusion among readers. Please follow the weakness part to provide more clear explanations.

3. **Strengthen and Expand the Experimental Evaluation**

  Based on the weakness part, could you provide more numerical results on VLM task, and do more experiments under previous evaluation metrcis on image classfication task.

4. **Improve Formatting and Structural Consistency**

  The paper's formatting, such as line spacing between titles and sections, lacks consistency, which can detract from readability and professionalism.

---

> ### Author Response · Authors · 2024-11-22
>
> Thank you for your detailed review and constructive feedback. We appreciate the thorough analysis and would like to address each of your main concerns (common concerns including clarity of single layer update and numerical results on VLM are addressed above):
>
> **(Related work section)** We provided a concise summary and discussion of related works in the background section and introduction section. We focus on how our work differs from existing ones and addresses current limitations. Nevertheless, we agree that a more comprehensive review would be beneficial for readers and we have included them in Appendix A of the revised pdf due to space limitation.
>
> **(Optimal Results Visualization)** We have highlighted the best results in Table 1 of the revised pdf as suggested.
>
> **(Evaluation Metrics Consistency)** We would like to clarify that our metrics are generally consistent with [1] but not all metrics are applicable to our setting. In our paper, we focus on 3 main metrics, including unlearning effectiveness, utility retention, and computation efficiency. These metrics are also measured by [1] via Unlearning Accuracy (UA), Remaining Accuracy (RA), Testing Accuracy (TA), Membership Inference Attack (MIA), and RTE. Specifically, their UA and RA measure the classification rate of the forgotten and remaining classes on the training set. Since our paper uses CLIP-based zero-shot classification, we do not perform classification on the training set, but instead test on unseen samples, thus their UA and RA are not applicable to us, and we used forget accuracy (FA) on unseen samples within the forget category. Also the MIA metric used in [1] is not directly applicable to our zero-shot classification setting, and MIA is not effective in identifying whether a sample is in the training set or not. For utility retention, we use test accuracy on CelebA (TA_CA) and ImageNet (TA_IN), which reflects the model performance on unrelated tasks.
>
> [1] Liu et al. Model Sparsity Can Simplify Machine Unlearning. NeurIPS 2023
>
> [2] Reza et al. Membership Inference Attacks Against Machine Learning Models. IEEE SP 2017
>
>
> **(Limited Performance Advantages in Table 2)** We agree that our method does not achieve the best performance on every metric, but it is superior in efficiency than all other methods, and is competitive in effectiveness. There is a natural trade-off between unlearning accuracy, model utility retention, and unlearning efficiency. As shown in Table 2, we are the only method that achieves the best trade-off and does not significantly underperform any other methods in all metrics.
>
> **(Improve Formatting and Structural Consistency)** We have tried to improve the overall line space consistency in the revised pdf. Some figures are large and cause large white spaces in the text and (sub)section headings. We will try our best to fix them in the final version by rearranging figures. If you notice some additional inconsistencies, please let us know.
>
> Please let us know if you have any remaining questions, and we would be happy to address them.

---

> ### Author Response · Authors · 2024-12-03
> **We Hope Our Responses Address Your Concerns – A Friendly Reminder of the Discussion Deadline**
>
> Dear Reviewer eMTT,
>
> We greatly appreciate your time and effort in reviewing our work! In our earlier responses, we have carefully addressed your concerns by conducting the relevant experiments you highlighted and providing thorough explanations to clarify your questions about the paper.
>
> As the discussion period deadline approaches, we welcome any additional questions or feedback you may have. We would be delighted to continue the conversation and provide further clarification if needed.
>
> We look forward to hearing from you!
>
> Best,
> Authors

---

> ### Comment · Reviewer_eMTT · 2024-12-03
> **response to author's official comments**
>
> Dear all authors and the other reviewers,
>
> I greatly appreciate the author's effort in responding the questions.
>
> From my end, compared with previous saliency-based machine unlearning algorithms, I think the paper's spotlight will be the unlearning effect in the VLM tasks.
>
> Thanks authors' effort for providing the experimental results in General Response - Part 2. However, the experimental setup for this VLM unlearning part which is still not quite clear. For example, how to calculate the forget accuracy during the evaluation and the details. And I am also curious of how to employing the SLUG algorithm in the CLIP vision encoder of the LLaVA, during the pre-training stage or fine-tuning on where.
>
> Moreover, I could see that author try to use 'unlearned Elon Musk' and 'unlearned Taylor Swift' to demonstrate the unlearning algorithm's effectiveness.  But it could be seen as some specific samples, we could not draw the conclusion with only two classes' results. Though Model Utility could be tested during the experiment, the Forget Accuracy part is far from comprehensive. Based on that, I could also not have the conclusion that 'unlearning specific concepts in the CLIP vision model can directly influence the language model’s output.' which you mentioned in the paper.
>
> I really appreciate your effort. However, based on the above analysis, I think the innovation of the paper is limited, so I may still keep my rating.
>
> Thanks

---

> > ### Author Response · Authors · 2024-12-04
> > **Thank you for your response**
> >
> > We hope our following response can help you
> >
> > **(Innovation)**
> > > However, based on the above analysis, I think the innovation of the paper is limited
> >
> > Our innovation has been acknowledged by reviewers 7HTf and eMTT in their Strengths sections. Our method is well motivated by the research in hierarchical information representation and task arithmetics in weight space. As noted by reviewers ufXX and AC4y, we addressed two critical issues of existing approaches: over-unlearning and computational inefficiency. Our method’s versatility is demonstrated through experiments across multiple tasks and models (CLIP, Stable Diffusion, and VLM) - a breadth not previously shown in existing work. The VLM unlearning is one of our highlights (but not the only one), and we show good performance through both qualitative and quantitative experiments.
> >
> >
> > **(Further clarification on VLMs experiment setup)**
> >
> > > how to calculate the forget accuracy during the evaluation and the details
> >
> > Regarding the forget accuracy evaluation process (previously detailed at https://openreview.net/forum?id=3p4raemLAH&noteId=HiymrvhRSq)
> >
> > 1. We evaluate each targeted celebrity using 100 test images of their identity
> > 2. For each image, we query the VLM with the question: "What is the name of the person in the image?"
> > 3. The forget accuracy is calculated as:
> > $$\text{Forget Accuracy} = \frac{\text{Num. Correct Prediction on Unlearned Identity}}{\text{Total Num. Test Instances}}$$
> >
> >
> > > I am also curious of how to employing the SLUG algorithm in the CLIP vision encoder of the LLaVA, during the pre-training stage or fine-tuning on where.
> >
> > We apply the SLUG algorithm directly to the CLIP vision encoder of LLaVA. Specifically, we take the pre-trained LLaVA model and modify a single layer within its vision encoder; there is no “pre-training stage” or conventional “fine-tuning” involved in our approach.
> > Furthermore, all our experiments utilize pre-trained models as their starting point. Our unlearning method focuses on updating only a single layer of the network, ensuring both efficiency and minimal disruption to the pre-trained model. While unlearning during the fine-tuning stage can be seen as a potential interpretation of our method, it is distinct from the layer-level editing approach we propose.
> >
> > **(Experiments on more identities)**
> >
> > In our rebuttal, we focused on "Elon Musk" and "Taylor Swift" as test cases to maintain consistency with our qualitative results and provide timely responses. While our method can handle other identities, time and resource constraints during the rebuttal period prevented more extensive experimental validation.
> >
> > > Based on that, I could also not have the conclusion that 'unlearning specific concepts in the CLIP vision model can directly influence the language model’s output.' which you mentioned in the paper.
> >
> > We have already demonstrated through qualitative and quantitative results that by unlearning a single layer of VLM vision encoder, we can forget the target identity while preserving general utility.

---

### Official Review · Reviewer_7HTf · 2024-11-04

**Soundness:** 3
**Presentation:** 3
**Contribution:** 3
**Rating:** 8
**Confidence:** 4

**Summary:**

This paper proposes an innovative approach to the issue of machine unlearning, which involves removing the influence of specific data subsets from trained machine learning models without retraining from scratch.

**Strengths:**

1. The method introduces a novel approach to targeted unlearning by updating a single targeted layer using a one-time gradient computation, which is distinct from more common methods that require iterative model updates across multiple layers.

2. The paper presents two new metrics, layer importance and gradient alignment, to determine the optimal layer and gradient direction for unlearning, enhancing the targeted precision of the process.

3. The experiment was sufficient for me.

**Weaknesses:**

1. Table 2: Performance overview of different unlearning methods on UnlearnCanvas. in this table, My intuition is that there is a lack of variance experiments, that is, running multiple rounds to see the best and worst performance of the algorithm.

**Questions:**

1. in table2, Why are there no variance experiments to illustrate the stability of various metrics?

---

> ### Author Response · Authors · 2024-11-22
>
> Thank you very much for your thoughtful review and positive assessment of our work.
>
> **(Variance experiments)** Regarding your concern about variance experiments in Table 2. This table follows the standard setup of UnlearnCanvas [1] benchmark, which reports the average scores of each method on each metric over 24,000 images. We used the reported numbers from UnlearnCanvas, which does not provide the variances for other methods. We provide the variance of our method below, which shows that our method achieves high performance with small variance.
>
> Table R3: SLUG performance on UnlearnCanvas with standard deviation (updated in Table 2 of the revised pdf)
>
>
>
> |    Method   |             |                  |             | Effectiveness |                   |             |         |         | Efficiency |          |
> |:-----------:|:-----------:|:----------------:|:-----------:|:-------------:|:-----------------:|:-----------:|:-------:|:-------:|:----------:|:--------:|
> |             |             | Style Unlearning |             |               | Object Unlearning |             | FID (↓) |   Time  |   Memory   |  Storage |
> |             |    UA (↑)   |      IRA (↑)     |   CRA (↑)   |     UA (↑)    |      IRA (↑)      |   CRA (↑)   |         | (s) (↓) |  (GB) (↓)  | (GB) (↓) |
> | SLUG (Ours) | 86.29±1.79% |    84.59±1.63%   | 88.43±1.61% |  75.43±2.91%  |    77.50±2.60%    | 81.18±1.46% |  75.97  |    39   |   3.61   |   0.04   |
>
> [1] Zhang et al. UnlearnCanvas: Stylized Image Dataset for Enhanced Machine Unlearning Evaluation in Diffusion Models. NeurIPS 2024
>
> Please let us know if you have any remaining questions, and we would be happy to address them.

---

### Author Response · Authors · 2024-11-22
**General Response - Part 2**

**(Quantitative results on VLM: eMTT, ufXX, AC4y)**
We performed additional experiments for quantitative evaluations of the VLM model (LLaVA-v1.5-7B) that we qualitatively analyzed in Figures 5 and 17 of our original submission. Specifically, we evaluate two instances of LLaVa-v1.5 unlearned for two targeted identities (Elon Musk and Taylor Swift) on three established VLM benchmarks: MME [1], GQA [2], and MMBench [3]. Higher scores on these benchmarks indicate better performance. The results are summarized in Table R2 below and Table 4 in the revised pdf.

Table R2: Quantitative evaluation on unlearning LLaVA-1.5 (added as Table 4 in the revised pdf)


|           Model          | Forget Accuracy (↓) |                   | VLM Benchmark Score (↑) |         |                  |
|:------------------------:|:-------------------:|:-----------------:|:-----------------------:|:-------:|:----------------:|
|                          |                     | MME [1] Cognition |    MME [1] Perception   | GQA [2] | MMBench [3] (en) |
|    Original LLaVA-1.5    |        99.50        |       323.57      |         1481.21         |  61.28  |       62.97      |
|   Unlearned “Elon Musk”  |         3.0         |       298.57      |         1354.61         |  60.70  |       61.34      |
| Unlearned “Taylor Swift” |         2.0         |       334.64      |         1336.09         |  60.72  |       60.14      |
|          Average         |         2.5         |       316.61      |         1345.35         |  60.71  |       60.74      |












The results in Table R2 highlight that SLUG achieves effective unlearning while maintaining utility, validating its effectiveness in the VLM context. For forget accuracy, we tested each targeted celebrity using 100 images associated with that identity. The forget accuracy evaluation involves the question, "What is the name of the person in the image?" with the corresponding celebrity name as the correct answer. The benchmark scores represent the utility of the model for vision-language tasks, which contains a broad set of coarse-to-fine-grained questions on visual recognition and visual reasoning. Overall, our results demonstrate that unlearned models accuracy on the targeted identity drops significantly, while benchmark scores remain high and close to those of the original model, preserving its overall utility. We have included this additional evaluation of VLMs in our manuscript and will make evaluation datasets publicly available.

[1] Fu et al. MME: A Comprehensive Evaluation Benchmark for Multimodal Large Language Models. arXiv:2306.1339

[2] Hudson et al. GQA: A New Dataset for Real-World Visual Reasoning and Compositional Question Answering. CVPR 2019

[3] Liu et al. MMBench: Is Your Multi-modal Model an All-around Player? NeurIPS 2024

**(Summary of manuscript PDF changes)**
1. Table 1. Best performing results are highlighted as suggested by eMTT
1. Table 2. Standard deviation added to the results as suggested by 7HTf
1. Added Appendix A. Related Work section as suggested by eMTT
1. Added Appendix B. Algorithm Pseudo Code for methodology clarification
1. Added Table 4. for experimental VLMs quantitative evaluation
1. Improve overall line space consistency as suggested by eMTT

---

### Author Response · Authors · 2024-11-22
**General Response - Part 1**

We thank all the reviewers for their thorough and constructive reviews of our paper. We greatly appreciate the detailed feedback and suggestions for improvement. Below we address the common concerns. We provide additional responses under individual comments.

**(Dataset curation: ufXX, AC4y)** We would like to clarify that we do not assume full access to the forget and retain data. In our experiments, we only use a tiny subset of the training data. Specifically, we follow a simple strategy of sampling a shard (containing around 7,900 images) from the LAION-400M (the CLIP's original training set) as the retain set. Through experiments, we observe consistent performance when using different shards, indicating that a single shard can adequately approximate the overall training data distribution. For the sake of simplicity, we use the first shard of LAION-400M, laion400m/00000.tar as the retain set. These settings can also be found in our code. For the forget set, we vary the number of images from 500 to 6,000, depending on the available images in the training set associated with the concept targeted for unlearning. This results in an $N_f/N_r$ ratio in the range of 0.06 to 0.76. The retain set constitutes only about 2.5 images per million of the original training images, which demonstrates that our method operates with minimal access to the full dataset while maintaining effectiveness.

**(Clarification of pareto plot and layer selection: eMTT, ufXX)** We balance Eqn (7) and (8) by identifying the pareto optimal set of layers that are not dominated by any other layers on both importance metric (7) and gradient alignment metric (8). As shown in Figure 2 (a), the colored dots are the layers on the pareto front, and other layers marked as “x”, are inferior to at least one layer on the pareto front in both (7) and (8). During unlearning, we only consider the layers on the pareto front. We iterate through all the layers on the pareto front (which is a small number, usually less than 10) and select the layer that provides best evaluation on the forget set. Note that inference time on the forget set is small and we can quickly test multiple layers.  For reference, we have included the pseudo code in Appendix B.

**(Clarification on single gradient: eMTT, ufXX,AC4y)** Our choice of updating the selected layer along a single gradient direction is motivated by the ideas of task arithmetic [1], which shows that a “task vector” that points from the pre-trained model weights to the fine-tuned model weights can be modified through arithmetic operations such as negation and addition to steer the behavior of the model. We extend this concept to perform unlearning arithmetic, where we start with the pre-trained model and use the gradient of the forget loss as the unlearning task vector. We showed through our experiments and analysis that updating the selected layer along a single gradient direction is effective. We conducted experiments using iterative gradients in Figure 2 (c) (f), and showed that iterative gradient calculation does not offer advantage over a single gradient and requires early stopping to prevent “over-unlearning” (i.e., utility loss). In addition, we performed an iterative gradient calculation on the single layer, and we show results as follows.

Table R1: Unlearning performance comparison between one-step update and iterative update (the experimental setup is consistent with Table 1 of the main paper)


|                      | FA@1 (↓)  | FA@5 (↓)  | TA_IN@1 (↑)  | TA_CA@1 (↑) |
|----------------------|-----------|-----------|--------------|-------------|
| “Iterative SLUG”     | 0.00      | 0.00      | 59.97        | 53.49       |
| SLUG (Ours, Table 1) | 0.00      | 0.00      | 59.96        | 58.32       |

Iterative gradient calculation and update of single layers achieves unlearning performance similar to single gradient direction (Table 1 in the main text) but slightly worse performance on CelebA classification (likely due to slight over-unlearning on identity). Iterative update also requires a careful learning rate tuning and early stopping; in contrast, single gradient direction offers an effective solution that is computationally simpler and robust to step size selection.

[1] Ilharco et al. Editing models with task arithmetic. ICLR 2023

---

### Author Response · Authors · 2024-11-26

Dear Reviewers,

We hope you have had an opportunity to review our response.

We have carefully addressed your concerns with detailed explanations and additional experiments.

Additionally, we have updated the paper PDF based on suggestions from all reviewers to enhance overall readability and improve the comprehensiveness of our experiments.

Please let us know if you have any further questions or concerns. We would be happy to address them and engage in further discussion.
We sincerely thank all the reviewers once again for your thoughtful comments and valuable feedback!

---

> ### Author Response · Authors · 2024-12-01
> **request for any outstanding questions/comments**
>
> Dear Reviewers,
>
> Please let us know if you have any further questions or concerns.
>
> We will be happy to address them and engage in further discussion.

---

### Public Comment · ~SeungBum_Ha1 · 2024-12-03

I am a Ph.D. student interested in machine unlearning. I found this research intriguing and have thoroughly read the paper. Additionally, I reproduced the experimental results using the code provided by the authors. The methodology is simple and intuitive, which I found very appealing. The key contribution of this paper lies in its ability to achieve unlearning of the target efficiently with a simple approach and low computational cost, even if the method is not entirely perfect. In addition, affecting all layers may cause serious generalization performance degradation.

Do you have experimental results that applied the same method across all (or some) layers? How bad is the generalization performance?

---

> ### Author Response · Authors · 2024-12-03
> **Thank you for your comment**
>
> Thank you for your interest in our work! We're glad to hear that you appreciate our method. Our results indicate that updating multiple or all layers can degrade generalization performance. Specifically, the top-1 test accuracy on CelebA decreases from 58.32% to 51.64% and 53.74%. For more details, please refer to our response here: https://openreview.net/forum?id=3p4raemLAH&noteId=KVuyjtAYgZ

---

> > ### Public Comment · ~SeungBum_Ha1 · 2024-12-03
> >
> > Thank you so much for sharing the experimental results.
> >
> > In my personal opinion, I think it would be a little better to solve the concern of reviewers by showing how it works on a larger or smaller retain set.

---

### Meta-Review · Area_Chair_cG9o · 2024-12-18

**Metareview:**

This submission received mixed reviews. Reviewer 7HTf provided the highest rating but did not engage in the post-rebuttal discussion despite reminders. The provided comments were also brief and lacked sufficient grounding to advocate for acceptance. The remaining reviewers maintained their reservations. Notably, Reviewer eMTT continued to express concerns about the innovation of the submission, even after considering the authors' rebuttal. Given these factors, I regretfully recommend rejection in its current form.

**Additional Comments On Reviewer Discussion:**

Reviewer 7HTf provided the highest rating but did not engage in the post-rebuttal discussion despite reminders. Reviewer eMTT maintained concerns about the innovation of the submission, even after considering the authors' rebuttal. The remaining reviewers, who assigned similar ratings of 5, remained inactive during the discussion phase. After carefully reviewing the reviewers' comments and the authors' response, I feel that this submission, while close, has not yet reached the acceptance bar.

---

### Decision · Program_Chairs · 2025-01-22

Reject